# Improved estimate of global gross primary production for reproducing its long-term variation, 1982-2017

Yi Zheng[1], Ruoque Shen[1], Yawen Wang[1,2], Xiangqian Li[1], Shuguang Liu[3], Shunlin Liang[4,5], Jing M. Chen[6,7], Weimin Ju[7,8], Li Zhang[9], Wenping Yuan[1,2*]

[1]School of Atmospheric Sciences, Sun Yat-sen University, Zhuhai 519082, Guangdong, China;
[2]Southern Marine Science and Engineering Guangdong Laboratory, Zhuhai 519000, Guangdong, China
[3]College of Life Science and Technology, Central South University of Forestry and Technology (CSUFT), Changsha, Hunan 410004, China
[4]Department of Geographical Sciences, University of Maryland, College Park, MD 20742 USA
[5]School of Remote Sensing Information Engineering, Wuhan University, Wuhan 430072, Hubei, China
[6]Department of Geography, University of Toronto, Canada, M5G 3G3
[7]International Institute for Earth System Sciences, Nanjing University, Nanjing, China.
[8]Jiangsu Center for Collaborative Innovation in Geographical Information Resource Development and Application, Nanjing, China.
[9]Key Laboratory of Digital Earth Science, Aerospace Information Research Institute, Chinese Academy of Sciences, Beijing 100094, China

*Correspondence to*: yuanwpcn@126.com (W. Yuan).

**Abstract.** Satellite-based models have been widely used to simulate vegetation gross primary production (GPP) at the site, regional, or global scales in recent years. However, accurately reproducing the interannual variations in GPP remains a major challenge, and the long-term changes in GPP remain highly uncertain. In this study, we generated a long-term global GPP dataset at 0.05° latitude by 0.05° longitude and 8-day interval by revising a light use efficiency model (i.e. EC-LUE model). In the revised EC-LUE model, we integrated the regulations of several major environmental variables: atmospheric $CO_2$ concentration, radiation components, and atmospheric vapor pressure deficit (VPD). These environmental variables showed substantial long-term changes, which could greatly impact the global vegetation productivity. Eddy covariance (EC) measurements at 95 towers from the FLUXNET2015 dataset, covering nine major ecosystem types around the globe, were used to calibrate and validate the model. In general, the revised EC-LUE model could effectively reproduce the spatial, seasonal, and annual variations in the tower estimated GPP at most sites. The revised EC-LUE model could explain 71% of the spatial variations in annual GPP over 95 sites. At more than 95% of the sites, the correlation coefficients ($R^2$) of seasonal changes between tower estimated and model simulated GPP are larger than 0.5. Particularly, the revised EC-LUE model improved the model performance in reproducing the interannual variations in GPP, and the averaged $R^2$ between annual mean tower estimated and model simulated GPP is 0.62 over all 55 sites with observations longer than 5-years, which is significantly higher than those of original EC-LUE model ($R^2 = 0.36$) and other LUE models ($R^2$ ranged from 0.06 to 0.30 with an average value of 0.16). At the global scale, GPP derived from light use efficiency models, machine learning models, and process-based biophysical models exist substantial differences in magnitude and interannual variations. The revised EC-LUE model quantified the mean global GPP from 1982 to 2017 as $106.2 \pm 2.9$ Pg C $yr^{-1}$ with the trend 0.15 Pg C $yr^{-1}$. Sensitivity analysis

indicated that GPP simulated by the revised EC-LUE model was sensitive to atmospheric $CO_2$ concentration, VPD, and radiation. Over the period of 1982–2017, the $CO_2$ fertilization effect on the global GPP (0.22 ±0.07 Pg C yr$^{-1}$) could be partly offset by increased VPD ($-0.17 \pm 0.06$ Pg C yr$^{-1}$). The long-term changes in the environmental variables could be well reflected in global GPP. Overall, the revised EC-LUE model is able to provide a reliable long-term estimate of global GPP. The GPP dataset is available at https://doi.org/10.6084/m9.figshare.8942336 (Zheng et al., 2019).

## 1 Introduction

Vegetation gross primary production (GPP) is the largest carbon flux component within terrestrial ecosystems and plays an essential role in regulating the global carbon cycle (Canadell et al., 2007; Zhao et al., 2010). As a primary variable of the terrestrial ecosystem cycle, GPP estimates will substantially determine other variables of the carbon cycle (Yuan et al., 2011). Satellite-based GPP models have been developed based on the light use efficiency (LUE) principle (Monteith, 1972; Potter et al., 1993; Running et al., 2004; Xiao et al., 2005; Yuan et al., 2007). Thus far, LUE models have been a major tool for investigating the spatio-temporal changes in GPP and the environmental dominates, either independently or by combining with other ecosystem models (Keenan et al., 2016; Smith et al., 2016).

However, current LUE models exhibit poor performance in reproducing the interannual variations in GPP. A previous study indicated that seven LUE models could only explain 6–36% of the interannual variations in GPP at 51 eddy covariance (EC) towers (Yuan et al., 2014). Similarly, a model comparison showed that none of the examined 16 process-based biophysical models or the 3 remote sensing products (BESS, MODIS C5, and MODIS C5.1) could consistently reproduce the observed interannual variations in GPP at 11 forest sites in North America (Keenan et al., 2012). Seven LUE models simulated the long-term trends in global GPP varied from $-0.15$ to 1.09 Pg C yr$^{-1}$ over the period 2000–2010 (Cai et al., 2014). An important reason for the poor performance in modeling the interannual variability is that the effect of environmental regulations on vegetation production is not completely integrated into the LUE models (Stocker et al., 2019). In particular, the long-term changes in several environmental variables are very important for accurately simulating the GPP series at the decadal scale. Several environmental variables should be included in GPP models. Firstly, as we all know the rising atmospheric $CO_2$ concentration in the past few decades substantially stimulated global vegetation growth (Zhu et al., 2016; Liu et al., 2017). Field experiments using greenhouses or open-top chambers showed that an increase of approximately 300 ppm in $CO_2$ concentration can increase the photosynthesis of C3 plants on the order of 60% (Norby et al., 1999). Free-air $CO_2$ enrichment (FACE) experiments generally confirmed the enhancement in net primary production (NPP) with the rising $CO_2$ concentration (Ainsworth and Long, 2005). For example, four FACE experiments indicated that the forest NPP consistently increased at the median of $23 \pm 2\%$ when the ambient $CO_2$ concentration was elevated to approximately 550 ppm (Norby et al., 2005). According to observations, the atmospheric $CO_2$ concentration has risen by approximately 20% from 340 ppm (1982) to 410 ppm (2018) (https://www.esrl.noaa.gov/). However, the effects of $CO_2$ fertilization on GPP have not been integrated in most current satellite-based LUE models.

Secondly, solar radiation, or more specifically the photosynthetic active radiation (PAR) substantially influences the vegetation production of the terrestrial ecosystem (Alton et al., 2007; Kanniah et al., 2012; Krupkova et al., 2017). Study indicated that the solar radiation incident at the earth surface underwent significant decadal variations (Wild et al., 2005). A comprehensive analysis based on the datasets of worldwide distributed sites indicated significant decreases in solar radiation (2% per decade) from the late 1950s to 1990 in the regions of Asia, Europe, North America, and Africa (Gilgen et al., 1998). A later assessment by Wild et al. (2005) showed that the radiation increased at widespread locations since the mid-1980s.

However, not only the total amount of solar radiation or PAR incident at the earth surface but more importantly, their partitioning into direct and diffuse radiations, impact the vegetation productivity (Urban et al., 2007; Kanniah et al., 2012). Increased proportion of diffuse radiation enhances vegetation photosynthesis, because a higher blue/red light ratio within the diffuse radiation may lead to higher light use efficiency (Gu et al., 2002; Alton et al., 2007). For example, the sharply increased diffuse radiation induced by the 1991 Mount Pinatubo eruption enhanced the noontime vegetation productivity of a deciduous forest in the next 2 years (Gu et al. 2003). Besides volcanic aerosols, clouds could also reduce the total and direct radiation, while increase the proportion of diffuse radiation. Yuan et al. (2010) found that the higher LUE at European forests than North America was because of the higher ratio of cloudy days in Europe. Yuan et al. (2014) further proved that the significantly underestimated GPP during cloudy days by six LUE models was because the effects of diffuse radiation on LUE were neglected in these models.

Thirdly, atmospheric vapor pressure deficit (VPD) is another factor that should be included in GPP models. As an important driver of atmospheric water demand for plants, VPD influences terrestrial ecosystem function and photosynthesis (Rawson et al., 1977; Yuan, et al., 2019). Rising air temperature increases the saturated vapor pressure at a rate of ~7%/K according to the Classius–Clapeyron relationship, and therefore, VPD will increase if the atmospheric water vapor content does not increase by exactly the same amount as the saturated vapor pressure. Numerous studies indicated significant changes in the relative humidity (ratio of actual water vapor pressure to saturated water vapor pressure) in both humid areas and continental areas located far from oceanic humidity (Van Wijngaarden and Vincent, 2004; Pierce et al., 2013). In particular, the global averaged land surface relative humidity decreased sharply after the late 1990s (Simmons et al. 2010; Willett et al. 2014) and the global averaged land surface VPD increased sharply after the late 1990s (Yuan, et al., 2019). The leaf and canopy photosynthetic rate declines when the atmospheric VPD increases due to stomatal closure (Fletcher et al., 2007). A recent study highlighted that increases in VPD rather than changes in precipitation would be a dominant influence on vegetation productivity (Konings et al., 2017). However, currently the influence of long-term VPD variations is not well expressed in many LUE models.

We have developed a LUE model, namely the EC-LUE model, by integrating remote sensing data and eddy covariance data to simulate daily GPP (Yuan et al., 2007; 2010). The model has been evaluated using the observations at EC towers located in Europe, North America, China, and East Asia, covering various ecosystem types (Yuan et al., 2007; 2010; Li et al., 2013). In this study, we revised the EC-LUE model by integrating the impacts of several environmental variables (i.e., atmospheric $CO_2$ concentration, radiation components, and atmospheric VPD) across a long-term temporal scale. Firstly, we evaluated the effectiveness of the revised EC-LUE model in determining the spatial, seasonal, and interannual variations in GPP from

multiple eddy covariance sites. Secondly, a global GPP dataset at 0.05° spatial resolution was generated based on the optimized model. Finally, we analyzed the contributions of the aforementioned environmental variables to the global GPP and discussed the spatial and interannual variations in GPP from different datasets.

## 2 Data and Methods

### 2.1 Data from the eddy covariance towers

The FLUXNET2015 dataset (http://www.fluxdata.org) includes over 200 variables of carbon fluxes, energy fluxes, and meteorological variables collected and processed at sites by the FLUXNET community. In our study, ninety-five EC sites in FLUXNET2015 dataset were utilized to optimize the parameters and evaluate the performance of the revised EC-LUE model, including nine major terrestrial ecosystem vegetation types (Table 1): evergreen broadleaf forests (EBF), evergreen needleleaf forests (ENF), deciduous broadleaf forests (DBF), mixed forests (MF), grasslands (GRA), savannas (SAV), shrubland (SHR), wetlands (WET), and croplands (CRO). More information about the characteristics of these sites can be referred to the FLUXNET website. For each site, the daily GPP, PAR, air temperature ($T_a$), and VPD were used in our study. The GPP variable (GPP_NT_VUT_REF) used in this study was estimated from night-time partitioning method. The corresponding net ecosystem exchange (NEE) was generated using variable friction velocity (USTAR) threshold for each year (VUT), in which 40 versions of NEE were created by using different percentiles of USTAR thresholds. The model efficiency between each version and the others 39 versions were calculated to test their similarities and the reference (REF) NEE was selected as the one with higher model efficiency sum (the most similar to the others 39). The daily meteorological 120 variables were gap-filled or downscaled from the ERA-interim reanalysis dataset in both space and time (Vuichard and Papale, 2015). The gap-filled technique of the carbon flux measurements and meteorological variables is the marginal distribution sampling (MDS) method described in Reichstein et al. (2005). In the FLUXNET dataset, there were quality flag ranged from 0 to 1 to indicate percentage of measured and good quality gap-filled data. For each variable, we used the daily/monthly values with more than 80% of good quality data (quality flag > 0.8). We aggregated the daily values to 8-day time step. And only the 8-day measurements with more than 5-day valid values were used.

<<Table 1>>

### 2.2 Data at the global scale

The global scale datasets used in this study are shown in Table 2. The meteorological reanalysis dataset was derived from the second Modern-Era Retrospective analysis for Research and Applications (MERRA-2) dataset. It was produced by NASA's Global Modeling and Assimilation Office that used an upgraded version of the GEOS-5 (Rienecker et al., 2011). It has been validated carefully using surface meteorological datasets and enhanced assimilation system to reduce the uncertainty in various meteorological variables globally. In our study, we obtained the daily mean air temperature ($T_a$, °C), mean dew point

temperature ($T_d$, °C), total direct PAR ($PAR_{dr}$, MJ m$^{-2}$ d$^{-1}$), and total diffuse PAR ($PAR_{df}$, MJ m$^{-2}$ d$^{-1}$) at 0.625° in longitude by 0.5° in latitude from 1982 to 2017. VPD was calculated from air temperature and dew point temperature:

$$SVP = 6.112 \times e^{\frac{17.67T_a}{T_a+243.5}} \tag{1}$$

$$RH = e^{\frac{17.625T_d}{T_d+243.04} - \frac{17.625T_a}{T_a+243.04}} \tag{2}$$

$$VPD = SVP \times (1 - RH) \tag{3}$$

where SVP is the saturated vapor pressure (k Pa), and RH is the relative humidity. We aggregated the daily variables (air temperature, VPD, direct PAR, and diffuse PAR) to 8-day interval temporal resolution. And these variables were resampled to the spatial resolution of 0.05° latitude by 0.05° longitude using the bilinear interpolation method.

The 8-day Global LAnd Surface Satellite-leaf area index (GLASS LAI) dataset at 0.05° latitude by 0.05° longitude was adopted to indicate vegetation growth from 1982 to 2017. It was produced using the general regression neural networks (GRNNs) trained with the fused MOD15 LAI and CYCLOPES LAI and the preprocessed MODIS/AVHRR reflectance data over the BELMANIP sites (Xiao et al., 2016). Products validation and comparison showed that the GLASS LAI product was spatially complete and temporally continuous with lower uncertainty (Xu et al., 2018).

Additionally, the MCD12Q1 product with IGBP classification scheme was used as land cover map. The ISLSCP II C4 Vegetation Percentage map was used to separate the C3 and C4 crop. The NOAA's Earth System Research Laboratory (ESRL) $CO_2$ concentration dataset was used to express the $CO_2$ fertilization effect.

<<Table 2>>

## 2.3 The revised EC-LUE model

The terrestrial vegetation GPP can be expressed as follows in the revised EC-LUE model:

$$GPP = (\varepsilon_{msu} \times APAR_{su} + \varepsilon_{msh} \times APAR_{sh}) \times C_s \times \min(T_s, W_s) \tag{4}$$

where $\varepsilon_{msu}$ is the maximum LUE of sunlit leaves; $APAR_{su}$ is the PAR absorbed by sunlit leaves; $\varepsilon_{msh}$ is the maximum LUE of shaded leaves; $APAR_{sh}$ is the PAR absorbed by shaded leaves; $C_s$, $T_s$, and $W_s$ represent the downward regulation scalars of atmospheric $CO_2$ concentration ($[CO_2]$), air temperature, and VPD on LUE ranging from 0 to 1; $min$ represents the minimum value.

The effect of atmospheric $CO_2$ concentration on GPP is determined by the following equations (Farquhar et al., 1980; Collatz et al., 1991):

$$C_s = \frac{C_i - \varphi}{C_i + 2\varphi} \tag{5}$$

$$C_i = C_a \times \chi \tag{6}$$

where $\varphi$ is the $CO_2$ compensation point in the absence of dark respiration (ppm); $C_i$ is the leaf internal $CO_2$ concentration; $C_a$ is the atmospheric $CO_2$ concentration; $\chi$ is the ratio of leaf internal to atmospheric $CO_2$ concentration which can be estimated as follows (Prentice et al., 2014; Keenan et al., 2016):

$$\chi = \frac{\varepsilon}{\varepsilon + \sqrt{VPD}} \tag{7}$$

$$\varepsilon = \sqrt{\frac{356.51K}{1.6\eta^*}} \tag{8}$$

where $\varepsilon$ is a parameter related to the 'carbon cost of water', which means the sensitivity of VPD to $\chi$; K is the Michaelis–Menten coefficient of Rubisco; $\eta^*$ is the viscosity of water relative to its value at 25 °C (Korson et al., 1969).

$$K = K_c\left(1 + \frac{P_0}{K_0}\right) \tag{9}$$

where $P_o$ is the partial pressure of $O_2$; $K_c$ and $K_o$ are the Michaelis–Menten constants for $CO_2$ and $O_2$ (Keenan et al., 2016):

$$K_c = 39.97 \times e^{\frac{79.43 \times (T_a - 298.15)}{298.15 \times R \times T_a}} \tag{10}$$

$$K_o = 27480 \times e^{\frac{36.38 \times (T_a - 298.15)}{298.15 \times R \times T_a}} \tag{11}$$

where $T_a$ is air temperature (unit: K); $R$ is the molar gas constant (8.314 J mol$^{-1}$ K$^{-1}$).

$T_s$ and $W_s$ can be expressed as follows:

$$T_s = \frac{(T_a - T_{min}) \times (T_a - T_{max})}{(T_a - T_{min}) \times (T_a - T_{max}) - (T_a - T_{opt}) \times (T_a - T_{opt})} \tag{12}$$

$$W_s = \frac{VPD_0}{VPD_0 + VPD} \tag{13}$$

where $T_{min}$, $T_{opt}$, and $T_{max}$ are the minimum, optimum, and maximum temperatures for vegetation photosynthesis, respectively (Yuan et al., 2007); $VPD_0$ is the half-saturation coefficient of the VPD constraint equation (k Pa).

$APAR_{su}$ and $APAR_{sh}$ can be expressed as follows (Chen et al., 1999):

$$APAR_{su} = \left(PAR_{dir} \times \frac{\cos(\beta)}{\cos(\theta)} + \frac{PAR_{dif} - PAR_{dif,u}}{LAI} + C\right) \times LAI_{su} \tag{14}$$

$$APAR_{sh} = \left(\frac{PAR_{dif} - PAR_{dif,u}}{LAI} + C\right) \times LAI_{sh} \tag{15}$$

$$PAR_{dif,u} = PAR_{dif} \times \exp\left(\frac{-0.5 \times \Omega \times LAI}{\cos(\bar{\theta})}\right) \tag{16}$$

where $PAR_{dir}$ is the direct PAR; $PAR_{dif}$ is the diffuse PAR; $PAR_{dif,u}$ is the diffuse PAR under the canopy; C represents the multiple scattering effects of direct radiation; $\Omega$ is the clumping index, which is set according to vegetation types (Tang et al., 2007); $\theta$ is the solar zenith angle; $\beta$ is the mean leaf–sun angle, which is set to 60°; $\bar{\theta}$ is the representative zenith angle for diffuse radiation transmission and can be expressed by LAI (Chen et al., 1999):

$$\cos(\bar{\theta}) = 0.537 + 0.025 \times LAI \tag{17}$$

The LAIs of shaded leaves ($LAI_{sh}$) and sunlit leaves ($LAI_{su}$) in Eqs. (14) and (15) are computed following Chen et al (1999):

$$LAI_{su} = 2 \times \cos(\theta) \times \left(1 - e^{-0.5 \times \Omega \times \frac{LAI}{\cos(\theta)}}\right) \tag{18}$$

$$LAI_{sh} = LAI - LAI_{su} \tag{19}$$

**2.4 Model calibration and validation**

Cross-validation method was used to calibrate and validate the revised EC-LUE model. Fifty percent of the sites were randomly selected to calibrate model parameters for each vegetation type, and the remaining 50% of the sites were used to validate the model. This parameterization process was repeated until all possible combinations of 50% sites were achieved for each vegetation type. The nonlinear regression procedure (Proc NLIN) in the Statistical Analysis System (SAS, SAS Institute Inc., Cary, NC, USA) was applied to optimize the model parameters ($\varepsilon_{msu}$, $\varepsilon_{msh}$, $\varphi$, and $VPD_0$) using 8-day estimated GPP based on

EC measurements. The mean GPP simulations of 8-day from all validation runs only were used to model validation. At global scale, mean calibrated parameter values (Table 3) were used to produce GPP dataset at $0.05° \times 0.05°$ spatial resolution and 8-day temporal resolution over 1982-2017. In order to investigate the uncertainties of the global GPP dataset, 10,000 sets of optimized parameters were randomly selected to simulate global GPP by assuming a normal distribution of these parameters (Table 3). The uncertainty of global GPP simulations was determined by the mean absolute deviation (MAD) of all the 10,000

simulations (Khair et al., 2017).

<<Table 3>>

Three metrics, the coefficient of determination ($R^2$), RMSE, and bias (the difference between observations and simulations) were adopted to evaluate the performance of the revised EC-LUE model. Additionally, Kendall's coefficient of rank correlation $\tau$ (Kanji, 1999) was used to quantify the agreement of seasonal changes between the simulated and tower estimated GPP. The

Kendall coefficient measured the tendency coherence between predicted and observed GPP by comparing the ranks assigned to successive pairs. If $GPP_{sim,j} - GPP_{sim,i}$ and $GPP_{obs,j} - GPP_{obs,i}$ have the same sign (positive or negative), the pair would be concordant, or discordant. A time-series data with $n$ observations, the Kendall's coefficient of rank correlation $\tau$ can be expressed:

$$\tau = \frac{C-D}{n(n-1)/2} \tag{20}$$

where $n(n-1)/2$ is the total combination of pairs, $C$ is the number of concordant pairs, and $D$ is the number of discordant pairs. The Kendall's coefficient ranged from -1 ($C = 0$) to 1 ($D = 0$). The Kendall's coefficient is much closer to 1, which means a stronger positive relationship between the seasonal patterns of the simulated and tower estimated GPP.

In addition, we compared the model performance of the revised EC-LUE model with seven light use efficiency models, three machine learning methods and ten process-based biophysical models. The participated light use efficiency models include

CASA (Potter et al., 1993), CFlux (Turner et al., 2006; King et al., 2011), CFix (Veroustraete et al., 2002), MODIS (Running et al., 2004), VPM (Xiao et al., 2005), VPRM (Mahadevan et al., 2008), and EC-LUE (Yuan et al., 2007). We calibrated the model parameters of all seven light use efficiency models based on the eddy covariance measurements using the same parameterization method as the revised EC-LUE model (see the above method), and then compared the GPP simulations of seven LUE models driven by EC tower-based meteorology data against the estimated GPP based on EC measurements. For

the comparison with machine learning models and process-based biophysical models, we collected their global monthly GPP products released by FLUXCOM (Jung et al., 2017) and TRENDY program (version 5) (Le Quéré et al., 2016), respectively.

FLUXCOM program uses the Artificial Neural Network method (FLUXCOM ANN), the Multivariate Adaptive Regression Splines method (FLUXCOM MARS), and the Random Forest method (FLUXCOM RF); and TRENDY program includes the CSIRO Atmosphere and Biosphere Land Exchange (CABLE) (Zhang et al., 2013), the coupled Canadian Land Surface Scheme and Canadian Terrestrial Ecosystem Model (CLASS-CTEM) (Melton and Arora, 2016), the Community Land Model (CLM) (Oleson et al., 2013), the Integrated Science Assessment Model (ISAM) (Jain et al., 2013), the land component of the Max Planck Institute Earth System Model (JSBACH) (Reick et al., 2013), the Joint UK Land Environment Simulator (JULES) (Clark et al., 2011), the Lund-Postdam-Jena General Ecosystem Simulator (LPJ-GUESS) (Smith et al., 2014), the Land surface Processes and eXchanges (LPX-Bern) (Stocker et al., 2014), the ORganizing Carbon and Hydrology In Dynamic EcosystEms (ORCHIDEE) (Krinner et al., 2005), and the Vegetation Integrated Simulator for Trace Gases (VISIT) (Kato et al., 2013). The monthly GPP simulations at all investigated EC sites were derived from their global products, and equally we obtained the monthly GPP simulations of the revised EC-LUE model from its global dataset driven by MERRA-2 reanalysis dataset.

## 2.5 Environmental contributions to long-term changes in GPP

To evaluate the contribution of the major environmental variables to GPP, including the atmospheric $CO_2$ concentration ([$CO_2$]), climate, and satellite-based LAI, two types of experimental simulations were performed. The first simulation experiment ($S_{ALL}$) was a normal model run, with all the environmental drivers changing over time. In the second type of simulation experiments ($S_{CLI0}$, $S_{LAI0}$, and $S_{CO20}$), two driving factors could be varied with time while maintaining the third constant at an initial baseline level. For example, the $S_{CLI0}$ simulation experiment allowed the LAI and atmospheric [$CO_2$] to vary with time while the climate variables were kept constant at 1982 values. The $S_{LAI0}$ ($S_{CO20}$) simulation experiments kept LAI (atmospheric [$CO_2$]) constant at 1982 values and varied the other two variables.

Considering the differences between the simulation results of the first type ($S_{ALL}$) and the second type ($S_{CO20}$ and $S_{LAI0}$) of experiments, the GPP sensitivities to atmospheric [$CO_2$] ($\beta_{CO2}$) and LAI ($\beta_{LAI}$) were estimated as follows:

$$\Delta GPP_{(S_{ALL}-S_{CO20})i} = \beta_{CO2} \times \Delta CO2_{(S_{ALL}-S_{CO20})i} + \varepsilon \tag{21}$$

$$\Delta GPP_{(S_{ALL}-S_{LAI0})i} = \beta_{LAI} \times \Delta LAI_{(S_{ALL}-S_{LAII0})i} + \varepsilon \tag{22}$$

where $\Delta GPP_i$, $\Delta CO2_i$, and $\Delta LAI_i$ denote the differences in the GPP simulations, atmospheric [$CO_2$], and LAI between the two model experiments from 1982 to 2017, and $\varepsilon$ is the stochastic error term.

The GPP sensitivities to the three climate variables: air temperature ($\beta_{Ta}$), VPD ($\beta_{VPD}$), and PAR ($\beta_{PAR}$) were calculated using a multiple regression approach:

$$\Delta GPP_{(S_{ALL}-S_{CLI0})i} = \beta_{Ta} \times \Delta Ta_{(S_{ALL}-S_{CLI0})i} + \beta_{VPD} \times \Delta VPD_{(S_{ALL}-S_{CLI0})i} + \beta_{PAR} \times \Delta PAR_{(S_{ALL}-S_{CLI0})i} + \varepsilon \tag{23}$$

where $\Delta Ta_i$, $\Delta VPD_i$, and $\Delta PAR_i$ denote the differences in Ta, VPD, and PAR time series between the two model experiments ($S_{ALL}$ and $S_{CLI0}$), respectively. The regression coefficient $\beta$ was estimated using the maximum likelihood analysis.

## 3 Results

### 3.1 Model performance

In general, the revised EC-LUE model could effectively reproduce the spatial, seasonal, and annual variations in the tower estimated GPP at most sites (Figs. 1–3). The revised EC-LUE model explained 71% and 64% of the spatial variations in GPP across all the validation sites by using the tower-derived meteorology data and the meteorological reanalysis dataset, respectively (Fig. 1).

<<Figure 1>>

The revised EC-LUE model also showed a good performance in reproducing the seasonal variations in GPP at most EC sites (Fig. 2). In this study, we compared the modeled and tower GPP at 8-day step for each site to examine the model capacity in reproducing the temporal variations of GPP. In terms of GPP simulations driven by tower-derived meteorology data, the coefficients of determination ($R^2$) varied from 0.26 at MY-PSO site to 0.96 at DK-Sor site, with most of them being statistically significant ($p$-value <0.05) (Fig. 2a), and the mean $R^2$ was 0.81 over all investigated sites. The low $R^2$ values (<0.4) were found

at three tropical forest sites (i.e., MY-PSO, BR-Sa1 and BR-Sa3). The averaged Kendall's correlation coefficient ($\tau$) was 0.63 over all sites, indicating a strong seasonal coherence between simulated and tower-estimated GPP (Fig. 2d). Similarly, $\tau$ at tropical forest sites were generally lower than other sites. According to the RMSE and absolute bias, the revised EC-LUE model performed very well at most sites. The averaged RMSE and absolute bias over all the sites were 2.13 and 0.81 g C m$^{-2}$ d$^{-1}$, respectively (Fig. 2b–c). In addition, there was no obvious difference between the seasonal GPP performances using the

tower-derived meteorology data and the meteorological reanalysis dataset (Fig. 2). On average, the revised EC-LUE model showed higher $R^2$ and $\tau$, and lower bias and RMSE than the original EC-LUE model (Fig. 2). Furthermore, we selected three sites with high $R^2$ (US-UMB; DBF; $R^2$ = 0.93), median $R^2$ (CN-Din; EBF; $R^2$ = 0.71), and low $R^2$ (Br-Sa3; EBF; $R^2$ = 0.39) to illustrate the time-series changes of observed/simulated GPP, LAI, and environmental factors (i.e., air temperature, VPD, and PAR) (Figs S1-S3). At US-UMB site, the model captured the GPP variations well all the year round with no obvious bias

(Fig. S1). At CN-Din site, the model generally performed well except the underestimation at the end of the year (November-December) with decreased LAI (Fig. S2). While, at Br-Sa3 site, the model could not capture the variations of GPP for the vegetation greenness and environmental factors varying slightly during the year (Fig. S3).

<< Figure 2>>

The ability of the LUE models to reproduce the interannual variations in GPP was investigated at 55 EC towers with

observations greater than 5-years (Table 1; Fig. 3). We examined the relations between the mean annual GPP simulations and observations at each site and used the coefficient correlation ($R^2$) and slope of the regression relationship to investigate the model capability in simulating the interannual variations in GPP. The result showed that the revised EC-LUE model could effectively determine the interannual variations in GPP (Fig. 3). Approximately 42% and 40% of the sites showed higher $R^2$ values (>0.5) by using the tower-derived meteorology data and the meteorological reanalysis dataset (Fig. 3a). The averaged

$R^2$ for the revised EC-LUE model was 0.44 by using the tower-derived meteorology data, which was significantly higher than

the original EC-LUE model ($R^2 = 0.36$) and other LUE models ($R^2$ ranged from 0.06 to 0.30 with an average value of 0.16) (Fig. 3c). The averaged $R^2$ for the revised EC-LUE model was 0.42 by using the meteorological reanalysis dataset. The averaged slopes of the revised EC-LUE model were 0.60 and 0.57 by using the tower-derived meteorology data and the meteorological reanalysis dataset (Fig. 3c).

<<Figure 3>>

Additionally, we examined the model performance of the revised EC-LUE model, other LUE models, machine learning models, and process-based biophysical models in TRENDY at monthly step by comparing against EC tower estimated GPP (Fig. 4). In comparison with seven LUE models driven by EC tower-based meteorology dataset, the overall $R^2$ of the revised EC-LUE model was 0.71, higher than the original EC-LUE model and other LUE models ($R^2$ ranged from 0.55 to 0.61) (Fig. 4a). For

each site, we compared the $R^2$/RMSE/absolute value of bias of the individual model with the averaged value of all the eight LUE models (each site has an averaged $R^2$/RMSE/absolute value of bias) (Fig. S4a1-c1). The revised EC-LUE model had higher $R^2$ than the mean $R^2$ of the eight LUE models at 62% sites, which was comparable with the original EC-LUE model (63% sites) and VPM model (60% sites) (Fig. S4a1). Moreover, the revised EC-LUE model showed the lower RMSE and bias compared to mean values of all eight LUE models at 68% and 67% sites respectively, which indicated the better performance

compared to the other LUE models at most sites (Fig. S4b1-c1). By using the global reanalysis meteorology data, we compared the performance of the revised EC-LUE model with three existing machine learning model products and ten process-based biophysical model products in TRENDY (Fig. 4b). The overall $R^2$ of the revised EC-LUE model ($R^2 = 0.57$) was higher than that of other models ($R^2$ ranged from 0.02 to 0.54) (Fig. 4b). The revised EC-LUE model, FLUXCOM ANN, and FLUXCOM MARS had more sites (over 90%) with higher $R^2$ than the mean $R^2$ (Fig. S4a2). And the revised EC-LUE model, FLUXCOM

MARS, and FLUXCOM RF showed the lower RMSE at more than 90% sites (Fig. S4b2). Compared to the other models, the revised EC-LUE model had highest site percentage (81%) with lower absolute value of bias (Fig. S4c2). Furthermore, the revised EC-LUE model had higher $R^2$, higher $\tau$, lower RMSE, and lower absolute value of bias at most of the sites (Fig. S5).

<<Figure 4>>

## 3.2 Spatial-temporal patterns of global GPP

A global GPP dataset at 0.05° latitude by 0.05° longitude and 8-day interval was generated from 1982 to 2017 based on the revised EC-LUE model. The global GPP was $106.2 \pm 2.9$ Pg C yr$^{-1}$ across the vegetated area averaged from 1982 to 2017. The GPP was high over the tropical forest areas, such as Amazon and Southeast Asia, where the moisture and temperature conditions are sufficient for photosynthesis (Fig. 5a). The GPP decreased with the decreasing gradients of temperature and precipitation (Fig. 5b). The moderate GPP was found in temperate and subhumid regions; and the lowest GPP was located in

arid or cold regions, where either precipitation or temperature is limited (Fig. 5b).

<<Figure 5>>

Long-term trend of GPP over the period of 1982–2017 was determined using a linear regression analysis (Fig. 6). In general, the revised EC-LUE model showed an increased trend in the annual mean GPP from 1982 to 2017. Approximately 69.5% of

the vegetated areas, mainly located in temperate and humid regions, showed increased trends. The spatial pattern of the GPP
trend along with the temperature and precipitation gradients was substantially heterogeneous (Fig. 6b). The decreased GPP
was found in the tropic regions, especially in the Amazon forest (Fig. 6a). The extremely cold or arid areas exhibited less
variations in GPP (Fig. 6b).

<<Figure 6>>

In addition, this study used the MAD of 10,000 simulations to quantify the uncertainty of estimated GPP globally (see methods).
Over the globe, the mean uncertainty of estimated GPP by the revised EC-LUE model is 19.33 Pg C yr$^{-1}$. The GPP uncertainties
were found to be low in high and middle latitudes, but relatively high in tropical forests (about 600 g C m$^{-2}$ yr$^{-1}$) (Fig. 7).

<<Figure 7>>

### 3.3 Contributions of environmental variables to GPP

To quantify the contributions of the environmental variables to long-term changes in GPP, we explored the sensitivity of global
summed GPP to climate variables (i.e., VPD, Ta, and PAR), LAI, and atmospheric $CO_2$ (Fig. 8). The global summed GPP
generated from different experimental simulations (section 2.5) exhibited differently in terms of the annual mean value, trend,
and standard deviation (Fig. 8a). The normal simulated GPP ($S_{ALL}$ GPP, all the environmental drivers changing over time)
significantly increased at the rate of 0.15 Pg C yr$^{-1}$, while the increasing rate of $S_{CLI0}$ GPP (climate variables were kept constant
at 1982 values) was even greater (0.41 Pg C yr$^{-1}$). On the contrary, the $S_{LAI0}$ GPP (LAI was kept constant at 1982 values) and
the $S_{CO20}$ GPP (atmospheric [$CO_2$] was kept constant at 1982 values) showed insignificantly decreasing trend at the rate of -
0.04 Pg C yr$^{-1}$ and -0.07 Pg C yr$^{-1}$ (Fig. 8a). The GPP sensitivity analysis showed that the global GPP decreased by 6.67 ±5.04
Pg C with a 0.1 kPa increase in VPD, which was comparable to the increase in GPP with 0.1 unit greening of LAI (i.e., $\beta_{LAI}$ =
4.78 ±0.72 Pg C 0.1 unit$^{-1}$) or 100 MJ increase in PAR (i.e., $\beta_{PAR}$ = 5.73 ± 3.22 Pg C 100 MJ$^{-1}$) (Fig. 8b). The global GPP
increased by 12.31 ±0.61 Pg C with a 100 ppm$^{-1}$ rise of atmospheric [$CO_2$] (i.e., $\beta_{CO2}$ = 12.31 ±0.61 Pg C 100 ppm$^{-1}$). Over
the period of 1982–2017, the increased VPD resulted in the global GPP decreases of −0.17 ±0.06 Pg C yr$^{-1}$, which could partly
counteract the fertilization effect of CO2 (0.22 ± 0.07 Pg C yr$^{-1}$). The global GPP showed a decreased trend after 2001 due to
the joint effect of increased VPD and decreased PAR (Fig. 8c). While the increased trend of GPP before 2000 was affected by
the rising atmospheric [$CO_2$], greening of LAI, and increased PAR (Fig. 8c).

<<Figure 8>>

### 4 Discussion

### 4.1 Model accuracy analysis

Numerous studies have shown that most GPP models can reproduce the spatial changes in GPP but failed to reproduce the
temporal variations (Keenan et al., 2012; Yuan et al., 2014). Therefore, the capacity to reproduce realistic interannual variations
for a GPP model is significantly important. In our study, the revised EC-LUE model performed a higher accuracy in

reproducing the interannual variations in GPP than did the original EC-LUE model and other LUE models. Yuan et al. (2014) reported that the averaged slope of the regression relation between the mean annual GPP simulated by seven LUE models and the mean annual GPP estimated from EC tower ranged from 0.19 to 0.56 (Fig. 3c). While the revised EC-LUE model showed a higher slope of regression relation (0.60), which is much closer to 1 than that obtained from other LUE models (Fig. 3c). The VPM GPP showed less interannual variations across most biomes ($R^2$ <0.5), probably because of the insensitivity of the environmental stress factors at the interannual scale (Zhang et al., 2017). In contrast, 42% of the sites showed higher $R^2$ values (>0.5) for the revised EC-LUE model. The improvements of the revised EC-LUE model in reproducing interannual variations are owing to the integration of several important environmental drivers for vegetation production (i.e., atmospheric $CO_2$ concentration, radiation components, and VPD), which exhibited large variations and contributed significantly to vegetation production at interannual scale.

By integrating the atmospheric $CO_2$ concentration, the revised EC-LUE model suggested a $CO_2$ sensitivity ($\beta_{CO2}$) of 12.31 ± 0.61 Pg C per 100 ppm (Fig. 8b), which indicates an increase of 11.6% in GPP with a rise of 100 ppm in atmospheric [$CO_2$]. Our estimate is comparable to the observed response of NPP to the increased $CO_2$ in the FACE experiments (13% per 100 ppm) and estimates of other ecosystem models (5–20% per 100 ppm) (Piao et al., 2013). The elevated atmospheric $CO_2$ concentration substantially contributes to vegetation productivity.

The evaporation fraction (EF), namely the ratio of evapotranspiration (ET) to net radiation (Rn), was used to indicate the water stress on vegetation growth in the original EC-LUE model (Yuan et al., 2007; 2010). While the atmospheric VPD was used to indicate water stress to avoid the aggregated errors from ET simulations in the revised EC-LUE model. Physiologically, vegetation production is sensitive to both atmospheric VPD and soil moisture availability to roots. Several studies have reported highly consistent interannual variability of VPD and soil moisture (Zhou et al., 2019a, b). In addition, recent studies highlighted that the increase in VPD had a larger limitation to the surface conductance and evapotranspiration than soil moisture over short time scales in many biomes (Novick et al., 2016; Sulman et al., 2016). Other studies have also suggested substantial impacts of VPD on vegetation growth (de Cárcer et al., 2018; Ding et al., 2018), forest mortality (Williams et al., 2013), and crop yields (Lobell et al., 2014). It is increasingly important to integrate the atmospheric water constraint to the carbon and water flux modeling.

## 4.2 Comparison of global GPP products

Global and regional GPP estimates remain highly uncertain despite the substantial advances in remote sensing technology, ground observations, and theory of carbon flux modeling (Zheng et al., 2018; Ryu et al., 2019). At regional scale, we compared the annual mean GPP between the revised EC-LUE model and other models across the bioclimatic zones in the Köppen-Geiger climate classification map (Beck et al., 2018) (Fig. 9). The GPP of the revised EC-LUE model was comparable to the mean value of other models for each bioclimatic zone (Fig. 9a). The GPP of different models exhibited large discrepancies in tropical regions (Af/Am/Aw) (Fig. 9a). The correlations ($R^2$) of GPP across all the bioclimatic zones between the revised EC-LUE model and other models ranged from 0.73 (LPX-Bern) to 0.95 (FLUXCOM MARS, FLUXCOM RF) (Fig. 9b).

<<Figure 9>>

At global scale, our study showed large differences in the magnitude of global GPP estimated by various models varying from 92.7 to 168.7 Pg C yr$^{-1}$ (Figs. 10–11). The LUE models simulated the global GPP ranging from 92.7 to 133.7 Pg C yr$^{-1}$ (Fig. 11a1). Several machine learning approaches estimated the global GPP ranging from 111.0 to 144.2 Pg C yr$^{-1}$ (Fig. 11a2). A comparison of ten global terrestrial ecosystem models of TRENDY showed that the global GPP ranged from 107.8 to 154.9 Pg C yr$^{-1}$ (Fig. 11a3). The revised EC-LUE model quantified the mean global GPP from 1982 to 2017 as 106.2 ± 2.9 Pg C yr$^{-1}$. Other studies also support the conclusion that there are large uncertainties in the GPP estimates. By comparing diverse GPP models and products, Anav et al. (2015) reported that the global GPP ranged from 112 to 169 Pg C yr$^{-1}$. Seven satellite-based LUE models estimated the global GPP ranged from 95.1 to 139.7 Pg C yr$^{-1}$ over the period of 2000–2010 (Cai et al., 2014).

<<Figure 10>>

The interannual variability and trend in GPP also vary substantially with different models. This study showed that the interannual variability (standard deviation) ranged from 0.32 to 5.89 Pg C yr$^{-1}$, with the trends varying from −0.05 to 0.84 Pg C yr$^{-1}$ (Fig. 11). The biophysical models showed large interannual variability, with the standard deviation ranging from 1.38 to 5.89 Pg C yr$^{-1}$. The LUE models estimated the interannual variability varied from 1.30 to 3.13 Pg C yr$^{-1}$. In contrast, the machine learning models exhibited less interannual variability with standard deviation under 1.0 Pg C yr$^{-1}$. The interannual variability of the revised EC-LUE model was 2.9 Pg C yr$^{-1}$ (Figs. 11b1–b3). In general, the GPP interannual variability before the year 2000 year was greater than that after the year 2001 for most of the biophysical models and LUE models (Figs. 11b1–b3). Most GPP models showed an increased trend or insignificant trend during all valid years and before 2000. Similar to the standard deviation, the trends of machine learning models were less than other models. Compared with the other models, CLASS-CTEM and the revised EC-LUE model showed a significant decreasing trend after 2001 (Figs. 11c1–c3), probably because of the joint effect of increased VPD and decreased PAR (Fig. 8c).

<<Figure 11>>

## 4.3 Model uncertainty

The uncertainties of our GPP dataset were low in high and middle latitude areas but high in tropical areas (Fig. 7). This is consistent with the validations at site level that the revised EC-LUE model showed the lowest accuracy over the tropical evergreen broadleaf forest sites (Fig. 2). Similarly, other satellite-based models exhibited a large uncertainty in the GPP simulations over tropical forest areas (Ryu et al., 2011; Yuan et al., 2014). For example, MODIS GPP product (MOD17) underestimated GPP at high productivity sites over the tropical evergreen forests (de Almeida et al., 2018). Regarding the quality of satellite data, a high cloud cover exists over tropical regions, introducing large uncertainties to FAPAR/LAI and other vegetation indices (e.g., NDVI and EVI). As suggested by de Almeida et al. (2018), the lack of reliable MOD15 FAPAR data during January to April as a result of the cloudiness contamination could have substantially affected the seasonality of GPP estimates. Besides, the quality of satellite data can even affect the evaluation of the interannual variations in GPP. Using

MODIS EVI data, Saleska et al. (2007) reported a large-scale green up in the Amazon evergreen forests during the drought of 2005. However, an opposite conclusion was drawn when the cloud-contaminated data were excluded from the analysis (Samanta et al., 2010). In our study, a significant decrease of GPP was found in the Amazon evergreen forests, which may be resulted from the sharp increase in VPD after the late 1990s (Yuan et al., 2019). Studies using optical satellite data can be influenced by the cloudiness contamination. Recently studies using cloud free satellite-based microwave data also reported a carbon loss in tropic forest (Liu et al., 2015; Fan et al., 2019).

The latest study highlighted that the aggregate canopy phenology rather than the climate changes is the main cause of the seasonal changes in photosynthesis in evergreen broadleaf forests (Wu et al., 2016). In particular, the new leaf growing synchronously with dry season litterfall may shift the old canopy to be younger, which can explain the significant seasonal increase (~27%) in the ecosystem photosynthesis. Therefore, the vertical changes in leaf age and photosynthesis ability with canopy depth are important to simulate the seasonal variations in carbon flux in tropical forests (Wu et al., 2017). These leaf trait related parameters can be simulated from the narrow-band spectra of leaves (Serbin et al., 2012; Dechant et al., 2017). Nevertheless, because of the limitation in obtaining the large scale hyperspectral remote sensing data, regional or global estimation of these parameters are currently unavailable.

The revised EC-LUE model does not integrate the regulation of soil nitrogen content on vegetation production. Atmospheric nitrogen deposition has exhibited a large increasing trend in the past few decades because of the excessive fossil fuel combustion in the industrial and transportation sectors and the abuse of nitrogenous fertilizer in the agricultural practice (Galloway et al., 2004). And the global land atmospheric nitrogen deposition is expected to further increase dramatically from 25–40 Tg N yr$^{-1}$ in the 2000s to 60–100 Tg N yr$^{-1}$ in 2100 (Lamarque et al., 2005). A meta-analysis of worldwide nitrogen addition experiments found that nitrogen addition could have a significantly positive effect on vegetation productivity (Liu and Greaver, 2009). As most terrestrial ecosystems are nitrogen limited, quantifying the spatio-temporal distributions of vegetation nitrogen content at large scales is essential to improve the accuracy of carbon flux estimation. Several studies quantified the leaf nitrogen content by detecting the nitrogen absorption spectra from the narrow-band of hyperspectral data (Cho, 2007). However, leaf water, starch, lignin, and cellulose overlap with the absorption characters of nitrogen in the shortwave infrared bands, making it difficult to retrieve the nitrogen content (Kokaly and Clark, 1999). Additionally, canopy structures, background, and illumination/viewing geometry can further decrease the capacity to detect leaf nitrogen (Yoder and Pettigrew-Crosby, 1995; Knyazikhin et al., 2013). Advances in inversion and statistical models of leaf or canopy nitrogen have emerged (Asner et al., 2011; Dechant et al., 2017; Wang et al., 2018), but these methods require further evaluation over large regions and the global map of leaf or canopy nitrogen is not available yet.

Additionally, the uncertainty of the revised EC-LUE model may arise by scale mismatches between eddy covariance flux footprint and input dataset. The eddy covariance flux footprint is generally less than 3 km$^2$ and varies depending on the wind speed, wind direction and atmospheric stability (Tan et al., 2006). In our studies, the revised EC-LUE model was run at 0.05 degree (~5 km$^2$) spatial resolution. The uncertainty of simulated GPP introduced by the scale effect is inevitable but smaller than that introduced by the model structures, parameters or input datasets (Sjostrom et al., 2013; Zheng et al., 2018).

## 5 Data availability

The $0.05° \times 0.05°$ global GPP dataset for 1982-2017 is available at https://doi.org/10.6084/m9.figshare.8942336 (Zheng et al., 2019). The dataset is provided in hdf format at 8-day interval. The valid value is ranged from 0 to 3000, and the background filled value is 65535. The scale factor of the data is 0.01. Each hdf file represents an 8-day GPP at daily value (unit: g C m$^{-2}$ day$^{-1}$). To obtain the summation of each 8-day (or 5-day or 6-day) period, please multiply the GPP value by corresponding days (8 for the first 45 values, and 5 or 6 for the last value in a year).

## 6 Conclusion

In this study, we produced a long-term global GPP dataset by integrating several major long-term environmental variables into a light use efficiency model, including atmospheric $CO_2$ concentration, radiation components, and atmospheric water vapor pressure. These environmental variables showed substantial long-term changes and contributed significantly to vegetation production at interannual scale. The revised EC-LUE model performed well in simulating the spatial, seasonal, and interannual variations in GPP across the globe. Particularly, it has a unique superiority in reproducing the interannual variations in GPP ($R^2 = 0.44$) compared with the original EC-LUE model ($R^2 = 0.36$) and other LUE models ($R^2$ ranged from 0.06 to 0.30 with an average value of 0.16). The GPP dataset derived from the revised EC-LUE model provides an alternative and reliable estimate of global GPP at the long-term scale by integrating the important environmental variables.

**Author contributions.** W. Yuan and Y. Zheng designed the research, performed the analysis, and wrote the paper; R. Shen, Y. Wang, and X. Li performed the analysis; S. Liu, S. Liang, J. Chen, W. Ju, and L. Zhang edited and revised the manuscript.

**Competing interests.** The authors declare that they have no conflict of interest.

## Acknowledgements

This study was supported by National Key Basic Research Program of China (2016YFA0602701), Changjiang Young Scholars Programme of China (Q2016161), Training Project of Sun Yat-sen University (16lgjc53), Fok Ying Tung Education Foundation (151015), and Beijing Normal University Project (2015KJJCA14). The covariance data used in the study was acquired and shared by the FLUXNET community, including these networks: AmeriFlux, AfriFlux, AsiaFlux, CarboAfrica, CarboEuropeIP, CarboItaly, CarboMont, ChinaFlux, Fluxnet-Canada, GreenGrass, ICOS, KoFlux, LBA, NECC, OzFlux-TERN, TCOS-Siberia, and USCCC. The ERA-Interim reanalysis data are provided by ECMWF and processed by LSCE. The FLUXNET eddy covariance data processing and harmonization was carried out by the European Fluxes Database Cluster, AmeriFlux Management Project, and Fluxdata project of FLUXNET, with the support of CDIAC and ICOS Ecosystem Thematic Center, and the OzFlux, ChinaFlux and AsiaFlux offices.

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

**Tables**

**Table 1: Information on the eddy covariance (EC) sites used in this study.**

| Site Name | Latitude | Longitude | Vegetation Type | Study Period |
|---|---|---|---|---|
| *DE-Kli | 50.89ºN | 13.52ºE | CRO | 2004-2012 |
| DE-RuS | 50.87ºN | 6.45ºE | CRO | 2011-2012 |
| FI-Jok | 60.90ºN | 23.51ºE | CRO | 2001-2003 |
| *FR-Gri | 48.84ºN | 1.95ºE | CRO | 2005-2012 |
| *US-ARM | 36.61ºN | 97.49ºW | CRO | 2003-2012 |
| *US-Ne1 | 41.16ºN | 96.47ºW | CRO | 2001-2012 |
| *US-Ne2 | 41.16ºN | 96.47ºW | CRO | 2001-2012 |
| *US-Ne3 | 41.17ºN | 96.43ºW | CRO | 2001-2012 |
| CA-TPD | 42.64ºN | 80.56ºW | DBF | 2012 |
| *DE-Hai | 51.08ºN | 10.45ºE | DBF | 2000-2012 |
| *DK-Sor | 55.49ºN | 11.64ºE | DBF | 2001-2012 |
| *FR-Fon | 48.48ºN | 2.78ºE | DBF | 2005-2012 |
| IT-PT1 | 45.20ºN | 9.06ºE | DBF | 2002-2004 |
| *IT-Ro2 | 42.39ºN | 11.92ºE | DBF | 2002-2008; 2010-2012 |
| JP-MBF | 44.39ºN | 142.32ºE | DBF | 2004-2005 |
| *US-Ha1 | 42.54ºN | 72.17ºW | DBF | 1992-2012 |
| *US-MMS | 39.32ºN | 86.41ºW | DBF | 1999-2012 |
| *US-Oho | 41.55ºN | 83.84ºW | DBF | 2004-2012 |
| *US-UMB | 45.56ºN | 84.71ºW | DBF | 2000-2012 |
| *US-UMd | 45.56ºN | 84.70ºW | DBF | 2008-2012 |
| *US-WCr | 45.81ºN | 90.08ºW | DBF | 1999-2006; 2011-2012 |
| *BR-Sa1 | 2.86ºS | 54.96ºW | EBF | 2002-2005; 2008-2011 |
| BR-Sa3 | 3.02ºS | 54.97ºW | EBF | 2001-2003 |
| CN-Din | 23.17ºN | 112.54ºE | EBF | 2003-2005 |
| *FR-Pue | 43.74ºN | 3.60ºE | EBF | 2000-2012 |
| *GF-Guy | 5.28ºN | 52.92ºW | EBF | 2004-2012 |
| *MY-PSO | 2.97ºN | 102.31ºE | EBF | 2003-2009 |
| CA-NS1 | 55.88ºN | 98.48ºW | ENF | 2002-2005 |
| *CA-NS2 | 55.91ºN | 98.52ºW | ENF | 2001-2005 |
| CA-NS3 | 55.91ºN | 98.38ºW | ENF | 2002-2005 |
| CA-NS4 | 55.91ºN | 98.38ºW | ENF | 2003-2005 |
| *CA-NS5 | 55.86ºN | 98.49ºW | ENF | 2001-2005 |

| | | | | |
|---|---|---|---|---|
| *CA-Qfo | 49.69ºN | 74.34ºW | ENF | 2003-2010 |
| CA-SF1 | 54.49ºN | 105.82ºW | ENF | 2003-2006 |
| *CA-SF2 | 54.25ºN | 105.88ºW | ENF | 2001-2005 |
| *CA-TP1 | 42.66ºN | 80.56ºW | ENF | 2003-2012 |
| *CA-TP2 | 42.77ºN | 80.46ºW | ENF | 2003-2007 |
| *CA-TP3 | 42.71ºN | 80.35ºW | ENF | 2003-2012 |
| CN-Qia | 26.74ºN | 115.06ºE | ENF | 2003-2005 |
| *CZ-BK1 | 49.50ºN | 18.54ºE | ENF | 2004-2012 |
| DE-Lkb | 49.10ºN | 13.30ºE | ENF | 2009-2012 |
| *DE-Obe | 50.78ºN | 13.72ºE | ENF | 2008-2012 |
| *DE-Tha | 50.96ºN | 13.57ºE | ENF | 1996-2012 |
| *FI-Hyy | 61.85ºN | 24.30ºE | ENF | 1996-2012 |
| IT-La2 | 45.95ºN | 11.29ºE | ENF | 2001 |
| *IT-Lav | 45.96ºN | 11.28ºE | ENF | 2003-2012 |
| *IT-Ren | 46.59ºN | 11.43ºE | ENF | 1999-2012 |
| *IT-SRo | 43.73ºN | 10.28ºE | ENF | 2001-2012 |
| *NL-Loo | 52.17ºN | 5.74ºE | ENF | 1996-2012 |
| *RU-Fyo | 56.46ºN | 32.92ºE | ENF | 1998-2012 |
| *US-Blo | 38.90ºN | 120.63ºW | ENF | 1997-2007 |
| *US-Me2 | 44.45ºN | 121.56ºW | ENF | 2002-2012 |
| US-Me6 | 44.32ºN | 121.61ºW | ENF | 2011-2012 |
| *US-NR1 | 40.03ºN | 105.55ºW | ENF | 1999-2012 |
| *CH-Cha | 47.21ºN | 8.41ºE | GRA | 2006-2008; 2010-2012 |
| *CH-Fru | 47.12ºN | 8.54ºE | GRA | 2006-2008; 2010-2012 |
| *CH-Oe1 | 47.29ºN | 7.73ºE | GRA | 2002-2008 |
| CN-Cng | 44.59ºN | 123.51ºE | GRA | 2007-2010 |
| CN-Du2 | 42.05ºN | 116.28ºE | GRA | 2007-2008 |
| CN-HaM | 37.37ºN | 101.18ºE | GRA | 2002-2003 |
| *CZ-BK2 | 49.49ºN | 18.54ºE | GRA | 2006-2011 |
| *NL-Hor | 52.24ºN | 5.07ºE | GRA | 2004-2011 |
| RU-Ha1 | 54.73ºN | 90.00ºE | GRA | 2002-2004 |
| US-AR1 | 36.43ºN | 99.42ºW | GRA | 2009-2012 |
| US-AR2 | 36.64ºN | 99.60ºW | GRA | 2009-2012 |
| *US-Goo | 34.25ºN | 89.87ºW | GRA | 2002-2006 |
| *US-IB2 | 41.84ºN | 88.24ºW | GRA | 2005; 2007-2011 |
| *BE-Bra | 51.31ºN | 4.52ºE | MF | 1999-2002; 2004-2012 |
| *BE-Vie | 50.31ºN | 6.00ºE | MF | 1997-2012 |

| | | | | |
|---|---|---|---|---|
| *CA-Gro | 48.22°N | 82.16°W | MF | 2004-2012 |
| CN-Cha | 42.40°N | 128.10°E | MF | 2003-2005 |
| JP-SMF | 35.26°N | 137.08°E | MF | 2003-2006 |
| *US-PFa | 45.95°N | 90.27°W | MF | 1996-2012 |
| *US-Syv | 46.24°N | 89.35°W | MF | 2001-2006; 2012 |
| AU-Ade | 13.08°S | 131.12°E | SAV | 2007-2009 |
| AU-Cpr | 34.00°S | 140.59°E | SAV | 2011-2012 |
| *AU-DaS | 14.16°S | 131.39°E | SAV | 2008-2012 |
| AU-Dry | 15.26°S | 132.37°E | SAV | 2009-2012 |
| AU-RDF | 14.56°S | 132.48°E | SAV | 2011-2012 |
| SD-Dem | 13.28°N | 30.48°E | SAV | 2007-2009 |
| *US-Ton | 38.43°N | 120.97°W | SAV | 2001-2012 |
| ZA-Kru | 25.02°S | 31.50°E | SAV | 2009-2012 |
| CA-NS6 | 55.92°N | 98.96°W | SRH | 2002-2005 |
| CA-NS7 | 56.64°N | 99.95°W | SRH | 2003-2005 |
| *CA-SF3 | 54.09°N | 106.01°W | SRH | 2002-2006 |
| ES-LgS | 37.10°N | 2.97°W | SRH | 2007-2009 |
| US-KS2 | 28.61°N | 80.67°W | SRH | 2003-2006 |
| CN-Ha2 | 37.61°N | 101.33°E | WET | 2003-2005 |
| DE-Akm | 53.87°N | 13.68°E | WET | 2010-2012 |
| DE-SfN | 47.81°N | 11.33°E | WET | 2012 |
| DE-Spw | 51.89°N | 14.03°E | WET | 2010-2012 |
| RU-Che | 68.61°N | 161.34°E | WET | 2002-2004 |
| US-Ivo | 68.49°N | 155.75°W | WET | 2004-2007 |
| *US-Los | 46.08°N | 89.98°W | WET | 2001-2008; 2010 |
| US-WPT | 41.46°N | 83.00°W | WET | 2011-2012 |

* indicates the site used to investigate the interannual variations in GPP with observations greater than 5-years.

**Table 2: Input datasets used to drive the revised EC-LUE model.**

| Variable | Dataset/provider | Source |
|---|---|---|
| Air temperature | MERRA2 | https://gmao.gsfc.nasa.gov/reanalysis/MERRA-2/ |
| Dew point temperature | MERRA2 | https://gmao.gsfc.nasa.gov/reanalysis/MERRA-2/ |
| Direct PAR | MERRA2 | https://gmao.gsfc.nasa.gov/reanalysis/MERRA-2/ |
| Diffuse PAR | MERRA2 | https://gmao.gsfc.nasa.gov/reanalysis/MERRA-2/ |
| LAI | GLASS | http://www.glass.umd.edu/Download.html |
| Landcover map | MCD12Q1 | https://lpdaac.usgs.gov/products/mcd12q1v006/ |
| C4 crop percentage | ISLSCP II C4 Vegetation Percentage | https://doi.org/10.3334/ORNLDAAC/932 |
| $CO_2$ concentration | NOAA's Earth System Research Laboratory | www.esrl.noaa.gov/gmd/ccgg/trends/ |

**Table 3: Optimized parameters ($\varepsilon_{msu}$, $\varepsilon_{msh}$, $\varphi$, and $VPD_0$) of the revised EC-LUE model for different vegetation types.**

| Vegetation Types | DBF | ENF | EBF | MF | GRA | CRO-C3 | CRO-C4 | SAV | SHR | WET |
|---|---|---|---|---|---|---|---|---|---|---|
| $\varepsilon_{msu}$ (g C MJ$^{-1}$) | $1.28 \pm 0.36$ | $1.72 \pm 0.42$ | $1.67 \pm 0.85$ | $1.38 \pm 0.21$ | $1.16 \pm 0.15$ | $1.25 \pm 0.42$ | $2.46 \pm 0.78$ | $2.24 \pm 0.68$ | $1.21 \pm 0.25$ | $1.34 \pm 0.26$ |
| $\varepsilon_{msh}$ (g C MJ$^{-1}$) | $3.59 \pm 0.66$ | $3.87 \pm 0.58$ | $4.35 \pm 0.72$ | $3.29 \pm 0.63$ | $1.91 \pm 0.46$ | $2.46 \pm 0.52$ | $5.64 \pm 1.02$ | $4.26 \pm 0.95$ | $2.71 \pm 0.52$ | $2.62 \pm 0.49$ |
| $\varphi$ (ppm) | $32 \pm 8.25$ | $25 \pm 7.59$ | $20 \pm 6.36$ | $49 \pm 11.25$ | $57 \pm 11.85$ | $43 \pm 9.56$ | $54 \pm 15.36$ | $54 \pm 12.23$ | $34 \pm 7.59$ | $36 \pm 10.32$ |
| $VPD_0$ (k Pa) | $1.15 \pm 0.25$ | $1.34 \pm 0.26$ | $0.57 \pm 0.15$ | $0.62 \pm 0.14$ | $1.69 \pm 0.35$ | $1.02 \pm 0.19$ | $1.53 \pm 0.31$ | $1.65 \pm 0.26$ | $1.34 \pm 0.21$ | $0.62 \pm 0.12$ |

**Figures**

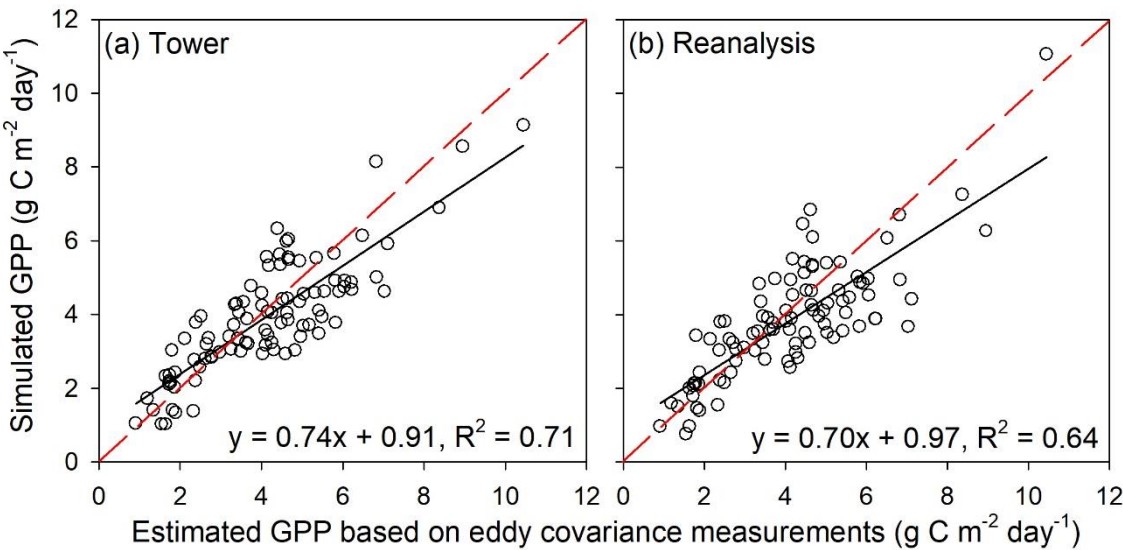

Figure 1: Comparisons between annual mean GPP estimated from EC towers and annual mean GPP simulated by the revised EC-LUE model. The modeled GPP were simulated using (a) tower-derived meteorology and (b) global reanalysis meteorology. The black lines are the regression lines, and the red dash lines are the 1:1 lines. The insert equations are the regression equations derived from all the sites.

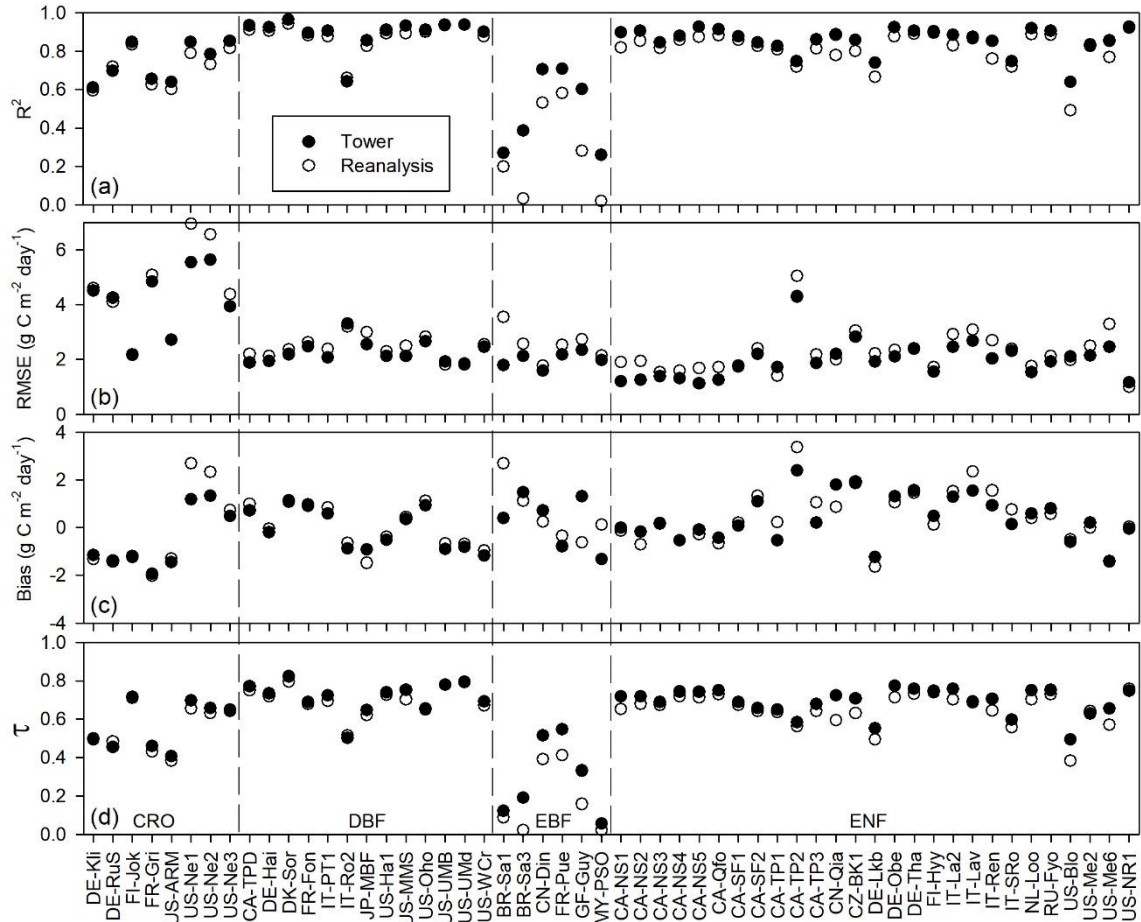

**Figure 2: Comparisons of 8-day mean GPP between the model simulated GPP and tower estimated GPP. Solid and open dots indicate the GPP simulations of the revised EC-LUE model derived from tower-derived meteorology data and meteorological reanalysis dataset, respectively; solid and open squares indicate the GPP simulations of the original EC-LUE model derived from tower-derived meteorology data and meteorological reanalysis dataset, respectively.**

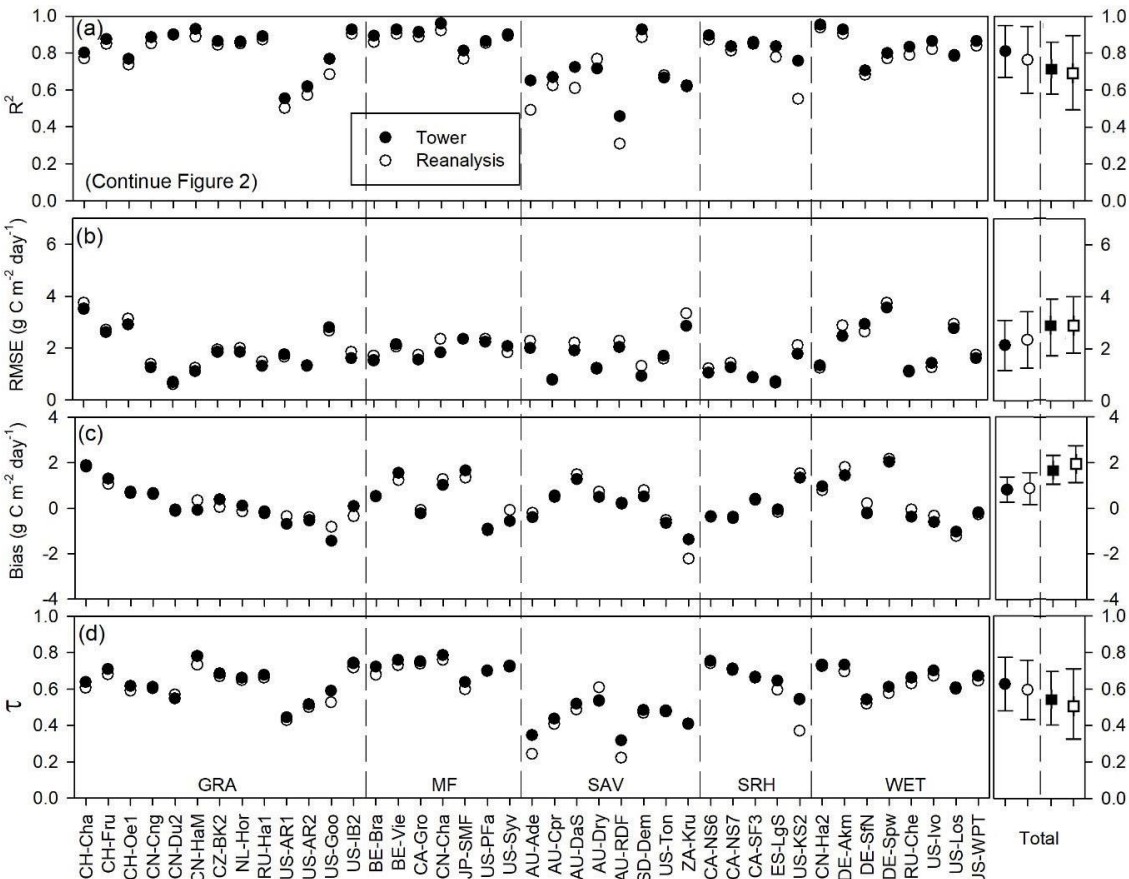


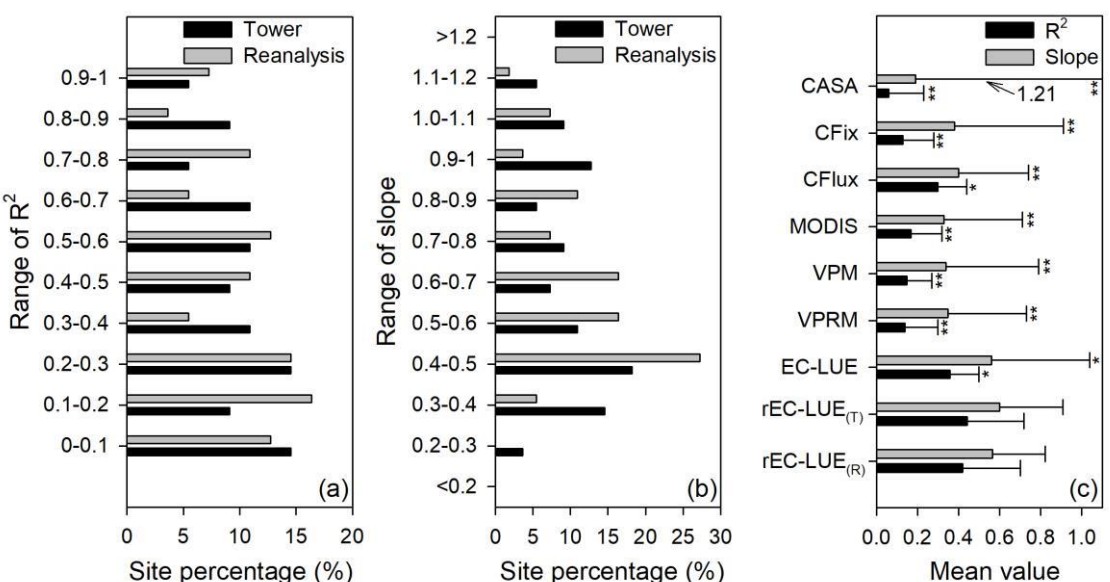

**Figure 3: Site percentage of (a) correlation coefficients (R$^2$), and (b) regression slopes between the model simulated and tower-based interannual variabilities in GPP. (c) Averaged values (error bars represent the standard deviations) of R$^2$ and slope for various LUE models. rEC-LUE$_{(T)}$ and rEC-LUE$_{(R)}$ indicate the revised EC-LUE models derived from tower-derived meteorology data and meteorological reanalysis dataset. The R$^2$ and slopes of the other seven LUE models (i.e., EC-LUE, VPRM, VPM, MODIS, CFlux, CFix, and CASA) in the figure were obtained from the study by Yuan et al. (2014). ** and * indicate a significant difference in statistic variables (R$^2$ and slope) between the rEC-LUE$_{(T)}$ and other LUE models (i.e., rEC-LUE$_{(T)}$ and other seven LUE models) at $p$-value < 0.01 and $p$-value<0.05, respectively.**

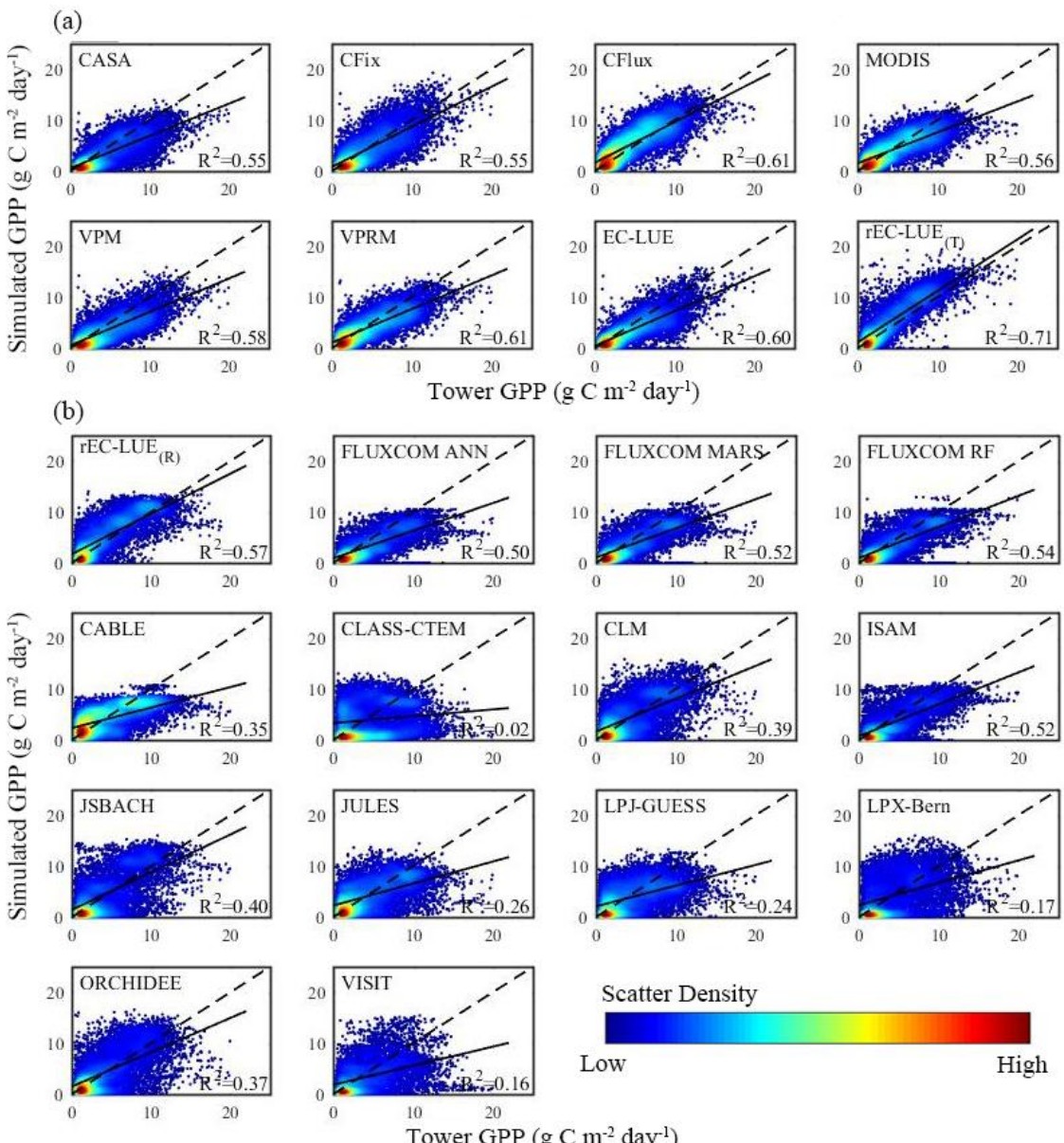

**Figure 4: Comparisons between estimated GPP based on EC measurements and GPP simulations in growing season (defined as temperature larger than 0°C) by the various models (including LUE models, machine learning models, and process-based biophysical models in TRENDY) at monthly scale. The comparison of GPP simulations were simulated using (a) tower-derived meteorology data and (b) global reanalysis meteorology data, respectively (see method 2.4). rEC-LUE(T) and rEC-LUE(R) indicate the simulations of the revised EC-LUE model derived from tower-derived meteorology data and global reanalysis meteorology data, respectively.**

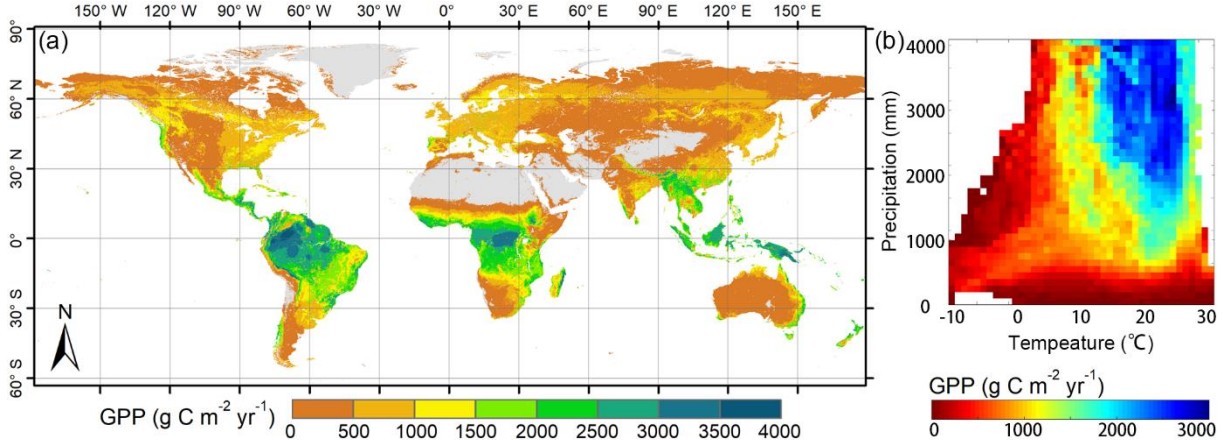

**Figure 5: Spatial pattern of global GPP simulated by the revised EC-LUE model during 1982–2017: (a) averaged annual GPP, (b) averaged annual GPP at different temperature and precipitation gradients.**

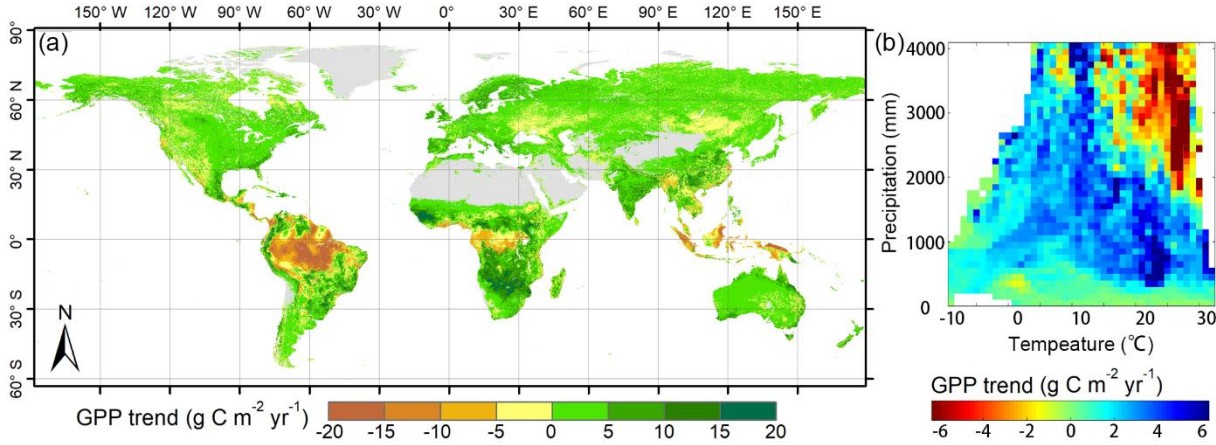

**Figure 6: Spatial pattern of global GPP trend simulated by the revised EC-LUE models during 1982–2017: (a) trend of annual GPP, (b) trend of annual GPP at different temperature and precipitation gradients.**

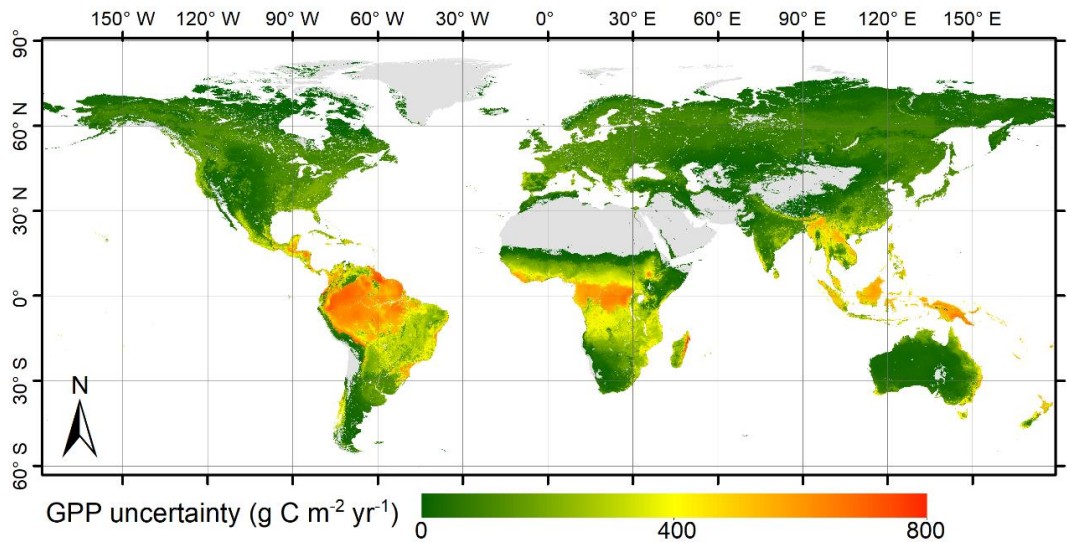

**Figure 7: Spatial pattern of the uncertainty in global GPP simulated by the revised EC-LUE model.**

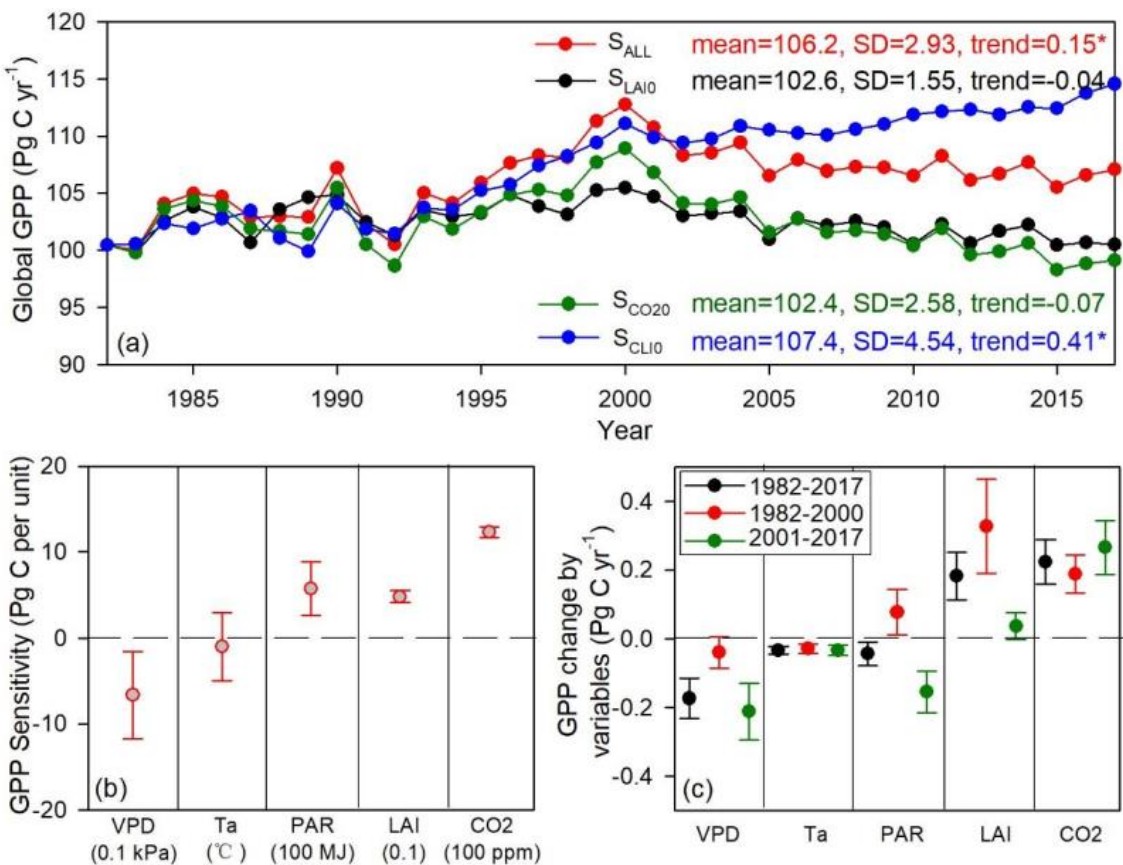

835

**Figure 8: Long-term changes in global GPP and the environmental regulations: (a) Global summed GPP derived from the four experimental simulations in section 2.5, (b) GPP sensitivity to climate variables (i.e., VPD, Ta, and PAR), LAI, and atmospheric $CO_2$, (c) contributions of climate variables (i.e., VPD, Ta, and PAR), LAI, and atmospheric $CO_2$ to GPP changes over 1982–2017, 1982–2000, and 2001–2017. \* indicates the significant level at *p*-value<0.05.**

840

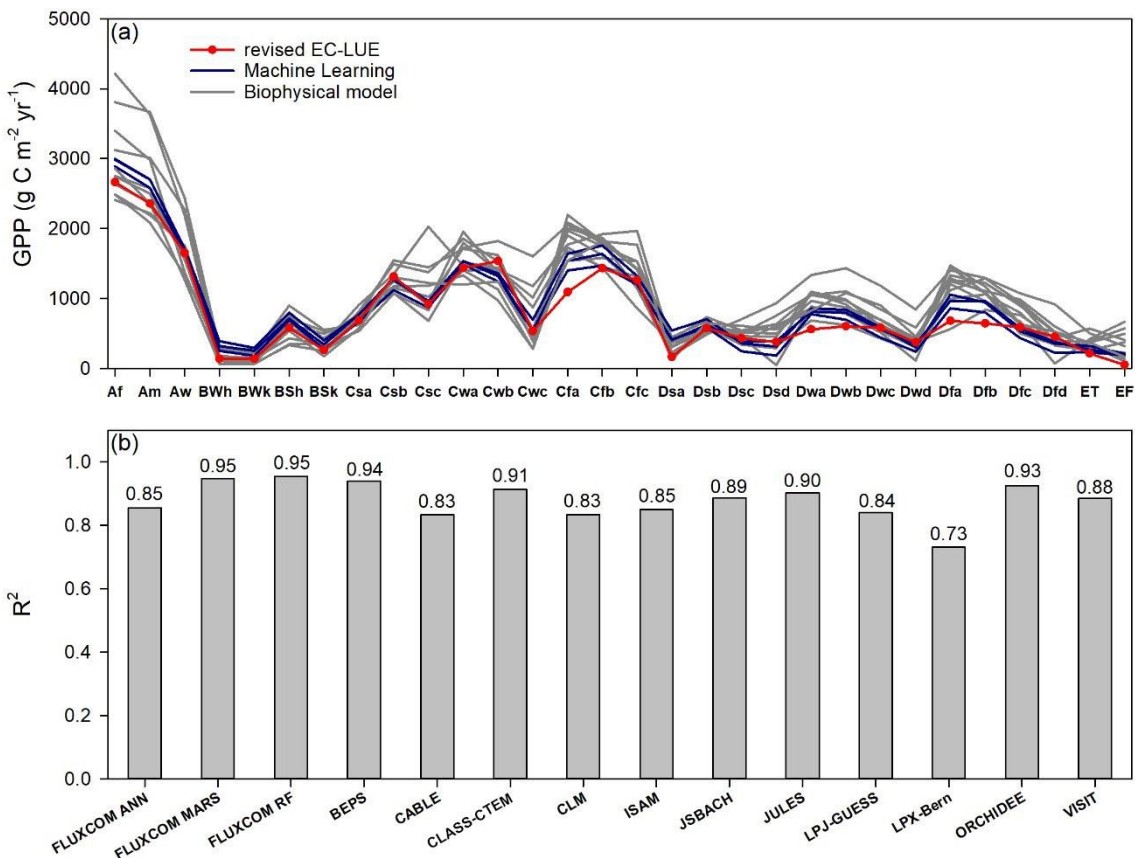

**Figure 9: Comparisons of long-term (1982 to 2010s) averaged GPP between the revised EC-LUE model and other models across bioclimatic zones in the Köppen-Geiger climate classification map (Beck et al., 2018). (a) the regional averaged value (b) correlation coefficients ($R^2$) of GPP across all the bioclimatic zones between the revised EC-LUE model and other models. These models including three machine learning models (FLUXCOM ANN, FLUXCOM MARS, FLUXCOM RF; Jung et al., 2017), biophysical models BEPS (Ju et al., 2006; Liu et al., 2018), and ten biophysical models in TRENDY (CABLE, CLASS-CTEM, CLM, ISAM, JSBACH, JULES, LPJ-GUESS, LPX-Bern, ORCHIDEE, and VISIT). The abbreviations for the bioclimatic zones are as follows: Af, tropical, rainforest; Am, tropical, monsoon; Aw, tropical, savannah; BWh, arid, desert, hot; BWk, arid, desert, cold; BSh, arid, steppe, hot; BSk, arid, steppe, cold; Csa, temperate, dry summer, hot summer; Csb, temperate, dry summer, warm summer; Csc, temperate, dry summer, cold summer; Cwa, temperate, dry winter, hot summer; Cwb, temperate, dry winter, warm summer; Cwc, temperate, dry winter, cold summer; Cfa, temperate, no dry season, hot summer; Cfb  temperate, no dry season, warm summer; Cfc, temperate, no dry season, cold summer; Dsa, cold, dry summer, hot summer; Dsb, cold, dry summer, warm summer; Dsc, cold, dry summer, cold summer; Dsd, cold, dry summer, very cold winter; Dwa, cold, dry winter, hot summer; Dwb, cold, dry winter, warm summer; Dwc, cold, dry winter, cold summer; Dwd, cold, dry winter, very cold winter; Dfa, cold, no dry season, hot summer; Dfb, cold, no dry season, warm summer; Dfc, cold, no dry season, cold summer; Dfd, cold, no dry season, very cold winter; ET, polar, tundra; EF, polar, frost.**

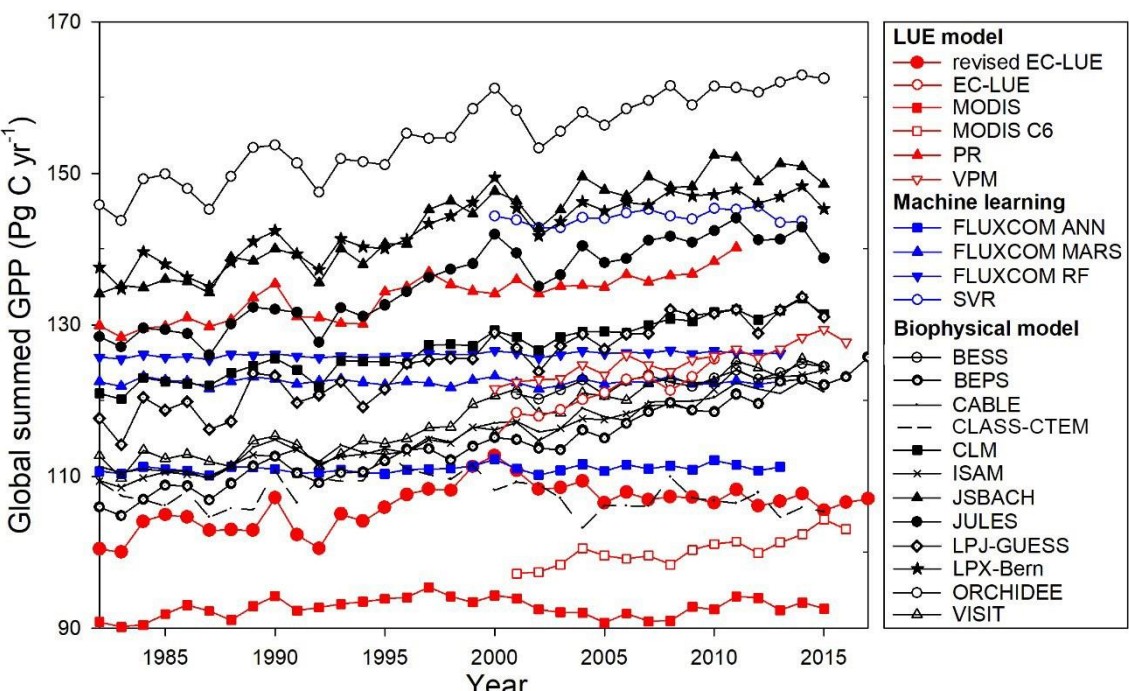

**Figure 10: Comparisons of annual global summed GPP estimates from various models. The datasets or model algorithms were obtained from: EC-LUE (Cai et al., 2014), MODIS (Smith et al., 2016), MOD17 C6 (Running et al., 2004), PR (Keenan et al., 2016), VPM (Zhang et al., 2017), FLUXCOM (Jung et al., 2017), SVR (Kondo et al., 2015), BESS (Jiang and Ryu, 2016), BEPS (Ju et al., 2006; Liu et al., 2018), and models in TRENDY (CABLE, CLASS-CTEM, CLM, ISAM, JSBACH, JULES, LPJ-GUESS, LPX-Bern, ORCHIDEE, and VISIT).**

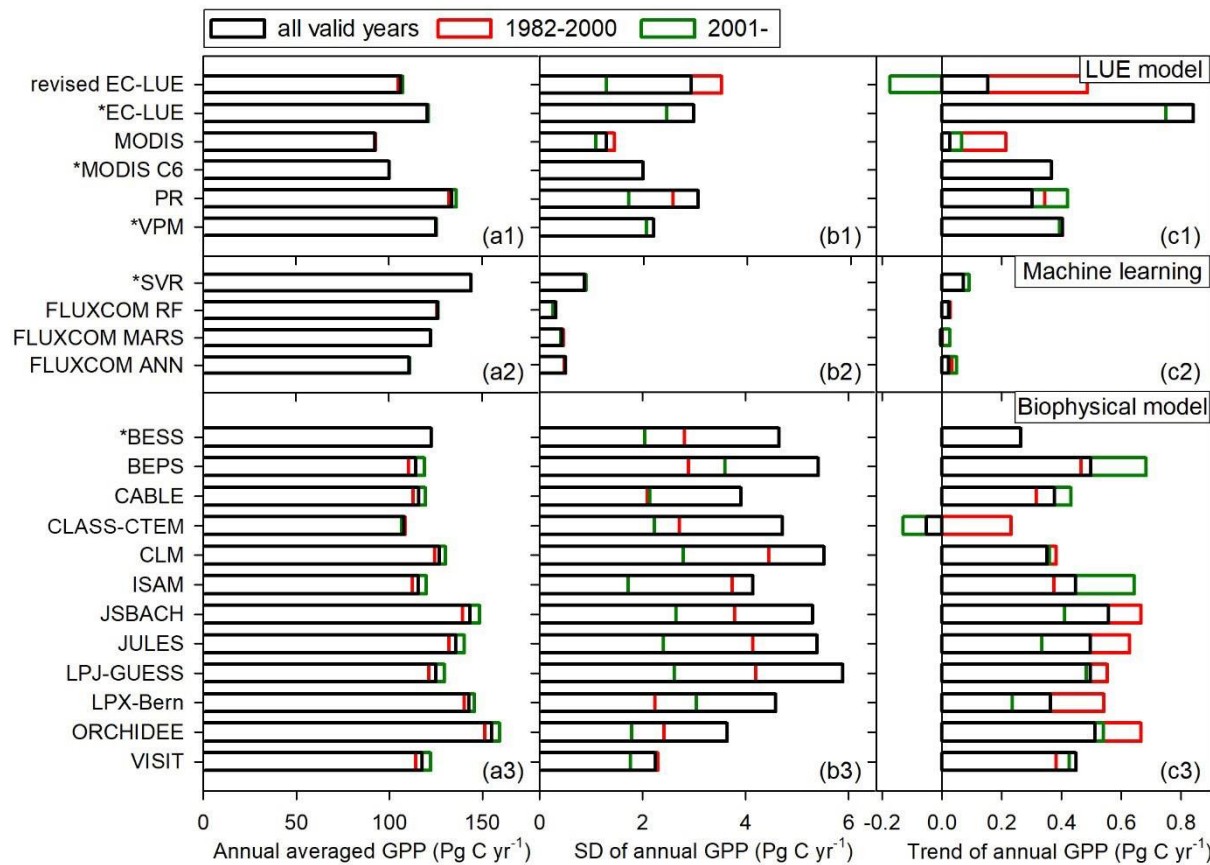

**Figure 11: Comparison of (a1)–(a3) averaged annual GPP, (b1)–(b3) interannual variability in annual GPP represented by standard deviation (SD), and (c1)–(c3) annual GPP trend among different GPP datasets or models. The references of these models are the same as in Figure 9. ∗ indicates that the valid period of the dataset begins from 2000 or 2001.**