# Peer review of "Improved estimate of global gross primary production for reproducing its long-term variation, 1982-2017"

_Earth System Science Data, 2019_

## Referee Comment (RC1) · Anonymous Referee #1 · 30 Sep 2019

LUE model is an important empirical model for estimating GPP. The authors added the impacts of CO2 concentration, diffuse/direct PAR, and VPD to the traditional LUE model, which showed improvement.

Line 18-35 In the abstract section, it is necessary to present some quantitative results that can directly prove the improvement of the revised EC-LUE model over other currently popular models.

Line 31-32 "The global GPP derived from different datasets exist substantial uncertainty in magnitude and interannual variations." Which datasets and which models were used here? Do the authors mean different datasets used to drive the revised

EC-LUE model? Or other models?

Line 448 Do the authors mean process based ecosystem models by biophysical models? And empirical or data-driven models by satellite-based models?

Line 50 The starting and ending years could be given while reporting a trend.

Line 70-90 (Major concern) The ratio of diffuse PAR is of course an important regulator of LUE for dense canopy. However, the amount of total PAR should not be ignored. LUE could rapidly decrease with the amount of total PAR because in clear sky the incident PAR could easily exceed light saturation point.

Section 2.1 At which temporal and spatial resolutions were the model run? And some the variables in the equations were not explained, e.g. epsilon in eq 4. Line 113 intercellular [CO2]? Line 111 add concentration after the second CO2. How was 356.51 in eq 5 determined?

Line 145-155 The fluxnet GPP contains many datasets of GPP according to the reference CO2 profile between sensor and canopy. Which dataset was used? And what is the temporal resolution of GPP, 30-min, daily, or 8-day?

Line 164 Daily mean air temperature?

Line 203-207 Those lines should go to method section.

Figure 4 (Major concern) Fig 4 could be expanded to better compare the performance of the revised EC-LUE model with other models in capturing the inter-annual and intra-annual GPP variations, to show the improvement of the revised EC-LUE model. This is important because there are a number of the existing models (process-based and the empirical LUE models as well the machine learning method). While the results about spatial and temporal variations of the GPP (from the new model and other models) should be compressed or even dropped.

[Figure]

2019.

---

## Referee Comment (RC2) · Anonymous Referee #2 · 10 Oct 2019

This paper reported some improvements of global gross primary production using the revised EC-LUE model. Overall, it lacks in detailed explanation and thorough validation to show the novelty of the proposed model if any. English must be significantly improved. Thus, I recommend rejecting the paper. Please see several major comments below.

1. Details are missing in many parts. Justification should follow when a decision or selection is done. For example, what is the rationale of dividing data into calibration and validation, and how was it done? How was the parameter optimization conducted? How was the collocation of different input data done? These are just a few of them.

[Figure]

Readers don't know what and how the authors exactly did, which limit the understanding of the proposed model and its evaluation.

2. Calibration vs. validation. As empirical models are dependent on data, a more robust approach should be adopted. Calibration sites were randomly selected? What if different calibration sites are used? A bootstrapping method might be adopted to see if consistent results can be achieved using different calibration data. Or n-fold cross validation would work. If consistent results were not obtained, the proposed model would be inherently unstable.

3. Seasonal analysis using time-series data should be conducted. Figs 2 and 3 are not sufficient to say that the proposed model showed a good performance in reproducing the seasonal variations in GPP as they don't contain any seasonal information. You may conduct statistical analysis by season, not simply based on stations.

4. More supporting references should follow in lines 250-251 if you want to say the decreased GPP was due to excessive precipitation and hot temp. In other words, both precipitation and temperature in Amazon significantly increased from 1982 to 2017? Seasonal factors might affect?

5. Scale issues should be carefully examined. Input data have different scales and the ground GPP measurements don't have the same scale with input data. What kinds of approaches were conducted when matching input data on the same spatial domain? How did the authors mitigate or consider the different scale issues between site GPP data and input variables?

6. Lines 282-285. Needs more uncertainty analysis by factor (e.g., radiation) to support this. Line 325. throughout the seasons? or different results by season? Again, seasonal analysis should be conducted.

7. Figure 8. Comparison by region (or continent) would make the paper robust. Are there any merits of using the proposed model in terms of the spatial domain?

8. Lines 371-372. Don't see any conclusive results to say that the model has a unique superiority in reproducing the inter annual variations in GPP at both site level and global scales. Superiority to what? Any comparison with other models (e.g., machine learning or physical models) to show the inter annual variations?

9. English needs to be carefully revised.

---

## Author Comment (AC1) · 18 Dec 2019

**Journal:** ESSD

**Title:** Improved estimate of global gross primary production for reproducing its long-term variation, 1982–2017

**MS No.:** essd-2019-126

**MS Type:** Data description paper

Dear editor and reviewer,

We are very grateful to your great efforts and constructive comments on our manuscript "Improved estimate of global gross primary production for reproducing its long-term variation, 1982–2017" (**MS No.:** essd-2019-126). The comments have helped improve the paper quite tremendously. We have carefully studied these comments and substantially revised our manuscript accordingly.

Here are our detailed responses to the comments point by point. Please note that the comments from the reviewer are in **bold** followed by our responses in regular text. The changes in our manuscript are underlined with red.

Please contact us if further materials or information are required. We deeply appreciate your consideration of our manuscript.

Sincerely,

Yi Zheng, Wenping Yuan, on behalf of all co-authors

School of Atmospheric Sciences,

Sun Yat-sen University, Zhuhai 519082, Guangdong, China

Email: yuanwpcn@126.com

**Response to Reviewer #1:**

**1. LUE model is an important empirical model for estimating GPP. The authors added the impacts of CO₂ concentration, diffuse/direct PAR, and VPD to the traditional LUE model, which showed improvement.**

Response: Thanks for your positive comments. We have revised the manuscript according to your comments point by point below.

**2. Line 18-35 In the abstract section, it is necessary to present some quantitative results that can directly prove the improvement of the revised EC-LUE model over other currently popular models.**

Response: As your suggestion, we adjusted and added more quantitative results to show the improvement of the revised EC-LUE model in the abstract section. The following is the revised abstract and the newly added sentences are underlined with red.

"Abstract. Satellite-based models have been widely used to simulate vegetation gross primary production (GPP) at the site, regional, or global scales in recent years. However, accurately reproducing the interannual variations in GPP remains a major challenge, and the long-term changes in GPP remain highly uncertain. In this study, we generated a long-term global GPP dataset at $0.05°$ latitude by $0.05°$ longitude and 8-day interval by revising a light use efficiency model (i.e. EC-LUE model). In the revised EC-LUE model, we integrated the regulations of several major environmental variables: atmospheric $CO_2$ concentration, radiation components, and atmospheric vapor pressure deficit (VPD). These environmental variables showed substantial long-term changes, which could greatly impact the global vegetation productivity. Eddy covariance (EC) measurements at 95 towers from the FLUXNET2015 dataset, covering nine major ecosystem types around the globe, were used to calibrate and validate the model. In general, the revised EC-LUE model could effectively reproduce the spatial, seasonal, and annual variations in the tower estimated GPP at most sites. The revised EC-LUE model could explain 71% of the spatial variations in annual GPP over 95 sites. At more than 95% of the sites, the correlation coefficients ($R^2$) of seasonal changes between tower estimated and model simulated GPP are larger than 0.5. Particularly, the revised EC-LUE model improved the model performance in reproducing the interannual variations in GPP, and the averaged $R^2$ between annual mean tower estimated and model simulated GPP is 0.44 over all 55 sites with observations longer than 5-years, which is significantly higher than those of original EC-LUE model ($R^2 = 0.36$) and other LUE models ($R^2$ ranged from 0.06 to 0.30 with an average value of 0.16). At the global scale, GPP derived from light use efficiency models, machine learning models, and process-based biophysical models exist substantial differences in magnitude and interannual variations. The revised EC-LUE model quantified the mean global GPP from 1982 to 2017 as $106.2 \pm 2.9$ Pg C yr$^{-1}$ with the trend 0.15 Pg C yr$^{-1}$. Sensitivity analysis indicated that GPP simulated by the revised EC-LUE model was sensitive to VPD, radiation, and $CO_2$ concentration. Over the period of 1982–2017, the $CO_2$ fertilization effect on the global GPP ($0.14 \pm 0.001$ Pg C yr$^{-1}$) could be offset by the effect of increased VPD ($-0.16 \pm 0.02$ Pg C yr$^{-1}$). The long-term changes in the environmental variables could be well reflected in global GPP. Overall, the revised EC-LUE model is able to provide a reliable long-term estimate of global GPP. The GPP dataset is available at https://doi.org/10.6084/m9.figshare.8942336 (Zheng et al., 2019)." (Line 18-40 in the revised manuscript)

**3. Line 31-32 "The global GPP derived from different datasets exist substantial uncertainty in magnitude and interannual variations." Which datasets and which models were used here? Do the**

**authors mean different datasets used to drive the revised EC-LUE model? Or other models?**

Response: We mean different GPP datasets simulated by other models in different studies. We modified the sentence as follow:

"At the global scale, GPP derived from light use efficiency models, machine learning models, and process-based biophysical models exist substantial differences in magnitude and interannual variations." (Line 33-34 in the revised manuscript)

**4. Line 48 Do the authors mean process based ecosystem models by biophysical models? And empirical or data-driven models by satellite-based models?**

**Line 48: Similarly, a model comparison showed that none of the examined 16 biophysical models nor the 3 satellite-based models could consistently reproduce the observed interannual variations in carbon exchange at 11 forest sites in North America (Keenan et al., 2012).**

Response: This sentence cited the results of Keenan et al., 2012, which includes 16 process-based biophysical models (i.e., BEPS, BIOME-BGC, Can-IBIS, CNCLASS, DLEM, ECOSYS, ED2, EDCM, ISAM, LoTEC-DA, LPJml, ORCHIDEE, SiB, SiB-CASA, SSiB2, and TECO) and 3 satellite-based model dataset (i.e., BESS, MODIS C5, and MODIS C5.1). BESS (Breathing Earth System Simulator) is a process-based model, and uses satellite-based leaf area index as driver. MODIS C5 and MODIS C5.1 indicate two MODIS-GPP products, and are based on MODIS-GPP algorithm which is satellite-based light use efficiency model. We changed the original sentences to make it clear:

"Similarly, a model comparison showed that none of the examined 16 process-based biophysical models or the 3 remote sensing products (BESS, MODIS C5, and MODIS C5.1) could consistently reproduce the observed interannual variations in GPP at 11 forest sites in North America (Keenan et al., 2012)." (Line 51-53 in the revised manuscript)

**5. Line 50 The starting and ending years could be given while reporting a trend.**

**Line 50: Seven LUE models simulated the long-term trends of global GPP varied −0.15 to 1.09 Pg C yr$^{-1}$ (Cai et al., 2014).**

Response: Thanks for your advice. We added starting and ending years as follow:

"Seven LUE models simulated the long-term trends of global GPP varied from −0.15 to 1.09 Pg C yr$^{-1}$ over the period 2000–2010 (Cai et al., 2014)." (Line 53-54 in the revised manuscript)

**6. Line 70-90 (Major concern) The ratio of diffuse PAR is of course an important regulator of LUE for dense canopy. However, the amount of total PAR should not be ignored. LUE could rapidly decrease with the amount of total PAR because in clear sky the incident PAR could easily exceed light saturation point.**

Response: Thanks for your deep thoughts. It is indeed that light saturation is an important response of GPP to varying PAR. The instantaneous LUE decreases rapidly when PAR exceed light saturation point. This is an instantaneous phenomenon which is obvious and nonnegligible at the hourly scale. The revised EC-LUE model was developed at the 8-day scale, and the light saturation can hardly be observed for the accumulation of GPP from hourly to 8-day temporal scale.

As an example, we examined the relation between GPP and PAR at hourly and 8-day scale at US-Ha1 site, respectively (Fig. R1). At hourly scale, there are obvious light saturation phenomenon when PAR exceeds 200 W m$^{-2}$ (Fig. R1a). However, at 8-day scale, the "ratio between GPP and LAI" (named GPP/LAI hereafter) keep increasing when PAR around its maximum value at 120 W m$^{-2}$ (Fig. R1b).

Some low GPP/LAI values may introduced by unfavorable climate conditions (e.g., low temperature or high VPD) or the uncertainty/error of the EC measurements. So we did not integrate the light saturation phenomenon in our current model.

[Figure]

Figure R1: Correlations of GPP and PAR at hourly and 8-day scale, taking US-Ha1 site as an example. At 8-day scale, we used the ratio between GPP and LAI to eliminate the influence of season patterns of LAI on GPP.

**7. Section 2.1 At which temporal and spatial resolutions were the model run? And some the variables in the equations were not explained, e.g. epsilon in eq 4. Line 113 intercellular [CO2]? Line 114 add concentration after the second CO₂. How was 356.51 in eq 5 determined?**
**Line 113-123: where $\varphi$ is the CO₂ compensation point in the absence of dark respiration (ppm); $C_i$ is the leaf internal CO₂ concentration; $C_a$ is the atmospheric CO₂ concentration; $\chi$ is the ratio of leaf internal to atmospheric CO₂ which can be estimated as follows (Prentice et al., 2014; Keenan et al., 2016):**

$$\chi = \frac{\varepsilon}{\varepsilon + \sqrt{VPD}} \tag{4}$$

$$\varepsilon = \sqrt{\frac{356.51 K}{1.6 \eta^*}} \tag{5}$$

$$K = K_c (1 + \frac{P_0}{K_0}) \tag{6}$$

$$K_c = 39.97 \times e^{\frac{79.43 \times (T-298.15)}{298.15RT}} \tag{7}$$

$$K_o = 27480 \times e^{\frac{36.38 \times (T-298.15)}{298.15RT}} \tag{8}$$

where $K_c$ and $K_o$ are the Michaelis–Menten constants for $CO_2$ and $O_2$; $P_o$ is the partial pressure of $O_2$; $Ta$ is air temperature (K); $\eta^*$ is the viscosity of water relative to its value at 25 ℃ depending on the air temperature (Korson et al., 1969); $R$ is the molar gas constant (8.314 J mol$^{-1}$ K$^{-1}$).

Response: The model was run at 8-day temporal resolution and 0.05 °×0.05 °spatial resolution. We added the information in the method section 2.4 (in the revised manuscript):

"Using the averaged value of the optimized parameters (Table 3), a global GPP dataset at 0.05 ° ×0.05 °spatial resolution and 8-day temporal resolution over 1982-2017 was produced." (Line 207-208 in the revised manuscript)

About the Eqs. (4)-(8) (in the original manuscript), we referred from Prentice et al. (2014) and Keenan et al. (2016). ε in Eq (4) is a parameter related to the 'carbon cost of water', which means the sensitivity of VPD to χ. We added the explanation of ε in the revised manuscript.

The 356.51 in Eq. (5) can be estimated using Eq (4)-(8) assuming the value of ε at 25℃ as 0.8 (T=298.15 K; VPD=1 kPa) described in Keenan et al. (2016), and we cited this paper.

In line 113 (in the original manuscript), we think the "leaf internal $CO_2$" and "intercellular $CO_2$" have a same meaning, so both are OK.

According to the response above, we modified this part as following:

"The effect of atmospheric $CO_2$ concentration on GPP is determined by the following equations (Farquhar et al., 1980; Collatz et al., 1991):

$$C_s = \frac{C_i - \varphi}{C_i + 2\varphi} \tag{5}$$

$$C_i = C_a \times \chi \tag{6}$$

where $\varphi$ is the $CO_2$ compensation point in the absence of dark respiration (ppm); $C_i$ is the leaf internal $CO_2$ concentration; $C_a$ is the atmospheric $CO_2$ concentration; χ is the ratio of leaf internal to atmospheric $CO_2$ concentration which can be estimated as follows (Prentice et al., 2014; Keenan et al., 2016):

$$\chi = \frac{\varepsilon}{\varepsilon + \sqrt{VPD}} \tag{7}$$

$$\varepsilon = \sqrt{\frac{356.51K}{1.6\eta^*}} \tag{8}$$

where ε is a parameter related to the 'carbon cost of water', which means the sensitivity of VPD to χ; K is the Michaelis–Menten coefficient of Rubisco; $\eta^*$ is the viscosity of water relative to its value at 25 ℃ (Korson et al., 1969).

$$K = K_c(1 + \frac{P_0}{K_0}) \tag{9}$$

where $P_o$ is the partial pressure of $O_2$; $K_c$ and $K_o$ are the Michaelis–Menten constants for $CO_2$ and $O_2$ (Keenan et al., 2016):

$$K_c = 39.97 \times e^{\frac{79.43 \times (T_a - 298.15)}{298.15 \times R \times T_a}} \tag{10}$$

$$K_o = 27480 \times e^{\frac{36.38 \times (T_a - 298.15)}{298.15 \times R \times T_a}} \tag{11}$$

where $T_a$ is air temperature (unit: K); $R$ is the molar gas constant (8.314 J mol$^{-1}$ K$^{-1}$)." (Line 155-170 in the revised manuscript)

**8. Line 145-155 The fluxnet GPP contains many datasets of GPP according to the reference CO$_2$ profile between sensor and canopy. Which dataset was used? And what is the temporal resolution of GPP, 30-min, daily, or 8-day?**

Response: In the FLUXNET2015 dataset, GPP was calculated considering flux portioning methods and friction velocity (USTAR) threshold. In our manuscript, we used the GPP variable GPP_NT_VUT_REF at daily temporal resolution in the FLUXNET2015 dataset. And, to match the temporal resolution of the remotely sensed LAI, we aggregated the daily GPP to 8-day temporal resolution. We modified the corresponding part to:

"The FLUXNET2015 dataset (http://www.fluxdata.org) includes over 200 variables of carbon fluxes, energy fluxes, and meteorological variables collected and processed at sites by the FLUXNET community. In our study, ninety-five EC sites in FLUXNET2015 dataset were utilized to optimize the parameters and evaluate the performance of the revised EC-LUE model, including nine major terrestrial ecosystem vegetation types (Table 1): evergreen broadleaf forests (EBF), evergreen needleleaf forests (ENF), deciduous broadleaf forests (DBF), mixed forests (MF), grasslands (GRA), savannas (SAV), shrubland (SHR), wetlands (WET), and croplands (CRO). More information about the characteristics of these sites can be referred to the FLUXNET website. For each site, the daily GPP, PAR, air temperature (Ta), and VPD were used in our study. The GPP variable (GPP_NT_VUT_REF) used in this study was estimated from night-time partitioning method. The corresponding net ecosystem exchange (NEE) was generated using friction velocity (USTAR) threshold for each year (VUT), in which 40 versions of NEE were created by using different percentiles of USTAR thresholds. The model efficiency between each version and the others 39 versions were calculated to test their similarities and the reference (REF) NEE was selected as the one with higher model efficiency sum (the most similar to the others 39). The daily meteorological variables were gap-filled or downscaled from the ERA-interim reanalysis dataset in both space and time (Vuichard and Papale, 2015). The gap-filled technique of the carbon flux measurements and meteorological variables is the marginal distribution sampling (MDS) method described in Reichstein et al. (2005). For each variable, we aggregated the daily values to 8-day time step. Only the 8-day measurements with more than 5-day valid values were used" (Line 108-123 in the revised manuscript)

**9. Line 164 Daily mean air temperature?**

**Line164-165: In our study, we obtained the daily air temperature (Ta, ℃), dew point temperature (Td, ℃), direct PAR, and diffuse PAR at 0.625° in longitude by 0.5° in latitude from 1982 to 2017.**

Response: Yes, modified.

"In our study, we obtained the daily mean air temperature (Ta, °C), mean dew point temperature (Td, °C), total PAR (PAR$_{dr}$, MJ m$^{-2}$ d$^{-1}$), and total diffuse PAR (PAR$_{df}$, MJ m$^{-2}$ d$^{-1}$) at 0.625 ° in longitude by 0.5 ° in latitude from 1982 to 2017." (Line130-132 in the revised manuscript)

**10. Line 203-207 Those lines should go to method section.**

**Line 203-207: This study used EC measurements at 42 sites to calibrate the parameter values and 43 sites to validate the model accuracy of the revised EC-LUE model. The parameters (ε$_{msu}$, ε$_{msh}$,**

**φ, and VPD₀) of each vegetation type are shown in Table 3. We evaluated the model performance by using the tower-derived meteorology data and global reanalysis meteorology, respectively. In general, the revised EC-LUE model could effectively reproduce the spatial, seasonal, and annual variations in the tower-estimated GPP at most of the calibration and validation sites (Figs. 1–4).**

Response: Yes, we agree that the contents about "calibration and validation" and "parameters" in these lines should be moved to method section. According to the suggestion of the second reviewer, we have used cross-validation method to estimate model parameters, and we rewrite this part and put them into method section "section 2.4 Model calibration and validation":

"Cross-validation method was used to calibrate and validate the revised EC-LUE model. Fifty percent of the sites were randomly selected to calibrate model parameters for each vegetation type, and the remaining 50% of the sites were used to validate the model. This parameterization process was repeated until all possible combinations of 50% sites were achieved for each vegetation type. The nonlinear regression procedure (Proc NLIN) in the Statistical Analysis System (SAS, SAS Institute Inc., Cary, NC, USA) was applied to optimize the model parameters ($\varepsilon_{msu}$, $\varepsilon_{msh}$, φ, and $VPD_0$) using 8-day estimated GPP based on EC measurements. The mean GPP simulations of 8-day from all validation runs only were used to model validation. Mean calibrated parameter values from all model runs were used to simulate GPP over the global scale (Table 3)." (Line 189-195 in the revised manuscript)

After consideration, we keep the result contents in these lines "In general, the revised EC-LUE model could effectively reproduce the spatial, seasonal, and annual variations in the tower-estimated GPP at most of the calibration and validation sites (Figs. 1–3)." in "section 3.1 Model performance".

**11. Figure 4 (Major concern) Fig 4 could be expanded to better compare the performance of the revised EC-LUE model with other models in capturing the inter-annual and intraannual GPP variations, to show the improvement of the revised EC-LUE model. This is important because there are a number of the existing models (process-based and the empirical LUE models as well the machine learning method). While the results about spatial and temporal variations of the GPP (from the new model and other models) should be compressed or even dropped.**

Response: Thank you. The main objective of our manuscript is focused on the improvement of the LUE models and produced a long-term GPP dataset. So, we compare the interannual variations of the revised EC-LUE model with other LUE models (CASA, CFix, CFlux, MODIS, VPM, VPRM, and the original EC-LUE model) as shown in Fig. 4. It is a good idea to compare with other kind models (process-based models and machine learning methods). However, we really appreciate the understanding that this comparison is quite beyond the field of this study. Moreover, due to large data gaps of measurements derived from eddy covariance towers, we need run process-based models at eddy covariance towers and obtain the corresponding simulations with observations of GPP in order to evaluate the model performance. This work needs contributions of model PIs, and which probably need take long time and great efforts. In addition, previous studies have provided the insights on this issue. Keenan et al. (2012) compared the performance of 16 process-based biophysical models and 3 satellite-based models (including the MODIS product) in reproducing the interannual variations in GPP. The result indicated the MODIS model performance was comparable to the process-based biophysical models. In our manuscript, the revised EC-LUE model (averaged $R^2 = 0.44$) was significantly better than the MODIS model (averaged $R^2 = 0.17$) at interannual scale. Therefore, we can conclude the similar result that our model was better than the process-based biophysical models compared in Keenan et al. (2012).

We rearranged the figures, Fig. 3 in the revised manuscript is the original Fig. 4.

[revised manuscript text omitted]

---

## Author Comment (AC2) · 18 Dec 2019

**Journal:** ESSD

**Title:** Improved estimate of global gross primary production for reproducing its long-term variation, 1982–2017

**MS No.:** essd-2019-126

**MS Type:** Data description paper

Dear editor and reviewer,

We are very grateful to your great efforts and constructive comments on our manuscript "Improved estimate of global gross primary production for reproducing its long-term variation, 1982–2017" (**MS No.:** essd-2019-126). The comments have helped improve the paper quite tremendously. We have carefully studied these comments and substantially revised our manuscript accordingly.

Here are our detailed responses to the comments point by point. Please note that the comments from the reviewer are in **bold** followed by our responses in regular text. The changes in our manuscript are underlined with red.

Please contact us if further materials or information are required. We deeply appreciate your consideration of our manuscript.

Sincerely,

Yi Zheng, Wenping Yuan, on behalf of all co-authors

School of Atmospheric Sciences,

Sun Yat-sen University, Zhuhai 519082, Guangdong, China

Email: yuanwpcn@126.com

**Response to Reviewer #2:**

**1. This paper reported some improvements of global gross primary production using the revised EC-LUE model. Overall, it lacks in detailed explanation and thorough validation to show the novelty of the proposed model if any. English must be significantly improved. Thus, I recommend rejecting the paper. Please see several major comments below.**

Response: Thanks for your deep thoughts and comments. The poor model performance in reproducing the interannual variability of GPP has been one of the most important uncertainties of satellite-based models, which will restrict our ability for quantifying the long-term trend of GPP over regional and global scales. This study aims to improve the model performance of a LUE model in reproducing interannual variability and produce a new long-term global GPP dataset. Meanwhile, we revised the manuscript according to your comments, and added detailed information on model parameterization and validation (please refer the following responses).

**2. Details are missing in many parts. Justification should follow when a decision or selection is done. For example, what is the rationale of dividing data into calibration and validation, and how was it done? How was the parameter optimization conducted? How was the collocation of different input data done? These are just a few of them. Readers don't know what and how the authors exactly did, which limit the understanding of the proposed model and its evaluation.**

Response: Sorry for confusion. We checked carefully the manuscript and made sure to represent the method, data and result clear. Here, the reviewer mentioned the model parameterization method, and we responded details in the next comment.

**3. Calibration vs. validation. As empirical models are dependent on data, a more robust approach should be adopted. Calibration sites were randomly selected? What if different calibration sites are used? A bootstrapping method might be adopted to see if consistent results can be achieved using different calibration data. Or n-fold cross validation would work. If consistent results were not obtained, the proposed model would be inherently unstable.**

Response: Thanks for your constructive comments. In the revised manuscript, we used cross validation method to calibrate model parameters. Cross validation method need more sites for each vegetation types, so we added the study sites from 84 to 95. We updated all the related method and result sections thoroughly (including all the tables and figures, and the related methods and results). And we produced and analyzed the global GPP datasets using the new parameters. Here we show the cross-validation method and the optimized parameter table. Other related modifications are too long to show here, please see them in the revised manuscript.

The detailed description on the cross-validation method is:

"Cross-validation method was used to calibrate and validate the revised EC-LUE model. Fifty percent of the sites were randomly selected to calibrate model parameters for each vegetation type, and the remaining 50% of the sites were used to validate the model. This parameterization process was repeated until all possible combinations of 50% sites were achieved for each vegetation type. The nonlinear regression procedure (Proc NLIN) in the Statistical Analysis System (SAS, SAS Institute Inc., Cary, NC, USA) was applied to optimize the model parameters ($\varepsilon_{msu}$, $\varepsilon_{msh}$, $\varphi$, and $VPD_0$) using 8-day estimated GPP based on EC measurements. The mean GPP simulations of 8-day from all validation runs only were used to model validation. Mean calibrated parameter values from all model runs were used to

simulate GPP over the global scale (Table 3)." (Line 189-195 in the revised manuscript)

The table of the optimized parameters are shown in Table 3:

Table 3: Optimized parameters ($\varepsilon_{msu}$, $\varepsilon_{msh}$, $\varphi$, and $VPD_0$) of the revised EC-LUE model for different vegetation types.

| Vegetation Types | DBF | ENF | EBF | MF | GRA | CRO-C3 | CRO-C4 | SAV | SHR | WET |
|---|---|---|---|---|---|---|---|---|---|---|
| $\varepsilon_{msu}$ (g C MJ$^{-1}$) | 1.28 ±0.36 | 1.72 ±0.42 | 1.67 ±0.85 | 1.38 ±0.21 | 1.16 ±0.15 | 1.25 ±0.42 | 2.46 ±0.78 | 2.24 ±0.68 | 1.21 ±0.25 | 1.34 ±0.26 |
| $\varepsilon_{msh}$ (g C MJ$^{-1}$) | 3.59 ±0.66 | 3.87 ±0.58 | 4.35 ±0.72 | 3.29 ±0.63 | 1.91 ±0.46 | 2.46 ±0.52 | 5.64 ±1.02 | 4.26 ±0.95 | 2.71 ±0.52 | 2.62 ±0.49 |
| $\varphi$ (ppm) | 32 ±8.25 | 25 ±7.59 | 20 ±6.36 | 49 ±11.25 | 57 ±11.85 | 43 ±9.56 | 54 ±15.36 | 54 ±12.23 | 34 ±7.59 | 36 ±10.32 |
| $VPD_0$ (kPa) | 1.15 ±0.25 | 1.34 ±0.26 | 0.57 ±0.15 | 0.62 ±0.14 | 1.69 ±0.35 | 1.02 ±0.19 | 1.53 ±0.31 | 1.65 ±0.26 | 1.34 ±0.21 | 0.62 ±0.12 |

Additionally, we examined the variability of model performance by using different combinations of calibration and validation sites (Fig. R1). We calculated the mean $R^2$ and RMSE across all validation sites for each combination, and used the coefficient of variation (CV) of $R^2$ and RMSE of all combinations to indicate the impacts of combinations on model performance. The averaged $R^2$ over all combinations ranged from 0.62 (EBF) to 0.88 (DBF) among various vegetation types, and the CV values of $R^2$ were mostly less than 0.11 (except EBF, CV = 0.32) (Fig. R1a-b). The averaged RMSE ranged from 1.33 g C m$^{-2}$ d$^{-1}$ (CRO-C3) to 5.84 g C m$^{-2}$ d$^{-1}$ (SRH) with CV varying from 0.06 to 0.30 (Fig. R1c-d). From statistics (mean, SD, and CV) of $R^2$ and RMSE, we can conclude our proposed model is robust with high accuracy.

[Figure]

Figure R1: Model performance for all the combinations of calibration and validation sites in cross-validation. (a) Averaged values of $R^2$ (error bars represent the standard deviation, namely SD), (b) coefficient of variation (CV) of $R^2$ (CV = SD/mean); (c) Averaged values of RMSE (error bars represent the SD), (b) CV of RMSE.

**4. Seasonal analysis using time-series data should be conducted. Figs 2 and 3 are not sufficient to say that the proposed model showed a good performance in reproducing the seasonal variations in GPP as they don't contain any seasonal information. You may conduct statistical analysis by season, not simply based on stations.**

[Figure]

**Figure 2: Comparisons of 8-day mean GPP between the observations at 42 calibration sites and the model simulations. Solid and open dots indicate the GPP simulations derived from tower-derived meteorology data and meteorological reanalysis dataset, respectively.**

[Figure]

**Figure 3: Comparisons of 8-day mean GPP between the observations at 43 validation sites and the model simulations. Solid and open dots indicate the GPP simulations derived from tower-derived meteorology data and meteorological reanalysis dataset, respectively.**

Response: In the revised manuscript we used cross-validation method, so we combined Figs. 2-3 to Fig. 2. In Figs. 2-3 (in the original manuscript), we calculated the correlation ($R^2$) between simulated and observed GPP at 8-day step for each site, and the correlation ($R^2$) indicates the consistence of temporal changes between GPP simulations and observations. We added these explanations as following:

"In Fig. 2, we compared the modelled GPP and tower GPP at 8-day step for each site to examine the capacity of our model in reproducing the seasonal variations." (Line 237-238 in the revised manuscript)

In addition, in the revised manuscript, we also added another index (Kendall's coefficient of rank correlation τ) to further quantify the agreement between the simulated and tower estimated GPP at seasonal patterns (Fig. 2d). We updated the Methods (Section 2.4 Model calibration and validation), Results (Section 3.1 Model performance), and Fig. 2d in the revise manuscript as following.

Methods (Section 2.4 Model calibration and validation):

"Additionally, Kendall's coefficient of rank correlation τ (Kanji, 1999) was used to quantify the agreement of seasonal changes between the simulated and tower estimated GPP. The Kendall coefficient measured the tendency coherence between predicted and observed GPP by comparing the ranks assigned to successive pairs. If $GPP_{sim,j} - GPP_{sim,i}$ and $GPP_{obs,j} - GPP_{obs,i}$ have the same sign (positive or negative), the pair would be concordant, or discordant. A time-series data with $n$ observations, the Kendall's coefficient of rank correlation τ can be expressed:

$$\tau = \frac{C-D}{n(n-1)/2} \qquad\qquad (20)$$

where $n(n-1)/2$ is the total combination of pairs, $C$ is the number of concordant pairs, and $D$ is the number of discordant pairs. The Kendall's coefficient ranged from -1 ($C = 0$) to 1 ($D = 0$). The Kendall's coefficient is much closer to 1, which means a stronger positive relationship between the seasonal patterns of the simulated and tower estimated GPP." (Line 197-206 in the revised manuscript)

Results (Section 3.1 Model performance):

"The averaged Kendall's correlation coefficient ($\tau$) was 0.63, indicating that the model simulated GPP had a strong seasonal coherence with tower estimated GPP. Similar to $R^2$, the lower Kendall's correlation coefficient ($\tau$) value sites were also located in the tropical forest areas." (Line 244-246 in the revised manuscript)

[Figure]

Figure 2: Comparisons of 8-day mean GPP between the model simulated GPP and tower estimated GPP. Solid and open dots indicate the GPP simulations derived from tower-derived meteorology data and meteorological reanalysis dataset, respectively.

Figure 2 (continue)

[Figure]

**5. More supporting references should follow in lines 250-251 if you want to say the decreased GPP was due to excessive precipitation and hot temp. In other words, both precipitation and temperature in Amazon significantly increased from 1982 to 2017? Seasonal factors might affect? Line 250-251: The decreased GPP areas were mainly distributed in the tropic regions with abundant precipitation and high temperature, particularly in the Amazon forest.**

Response: Sorry for confusion. It is not our purpose to say the abundant precipitation and high temperature is the cause of decreased GPP in tropic regions. We revised the sentence to:

"The decreased GPP was found in the tropic regions, especially in the Amazon forest." (Line 274 in the revised manuscript).

The decreased GPP in the tropic regions were mainly due to the suppression of the increased atmospheric water demand indicated by atmosphere vapor pressure deficit (VPD). We have reported the detailed cause of GPP decreases responded to the increased VPD in our recent paper (Yuan et al., 2019), therefore, we appreciate your understanding that we did not discuss the details. In addition, the main objective of this manuscript is to introduce the revised EC-LUE model and long-term global GPP dataset produced by EC-LUE model.

**6. Scale issues should be carefully examined. Input data have different scales and the ground GPP measurements don't have the same scale with input data. What kinds of approaches were conducted when matching input data on the same spatial domain? How did the authors mitigate or consider the different scale issues between site GPP data and input variables?**

Response: At global scale, the spatial resolution of satellite-based GLASS LAI dataset is 0.05 °latitude by 0.05 °longitude. We downscaled the meteorological reanalysis data (temperature, direct PAR, diffuse PAR, and VPD) to 0.05 °latitude by 0.05 °longitude using the bilinear interpolation method to match the spatial resolution of LAI. We have reported the detailed method in the manuscript:

"We aggregated the daily variables (air temperature, VPD, direct PAR, and diffuse PAR) to 8-day interval temporal resolution. And these variables were resampled to the spatial resolution of 0.05 ° latitude by 0.05 ° longitude using the bilinear interpolation method." (Line 136-138 in the revised manuscript)

At site level, we calibrated and validated the model using the tower observed meteorology data and global reanalysis meteorology data, respectively. The tower observed meteorology data were directly obtained from the measurement of FLUXNET and the global reanalysis meteorology data were extracted from the processed global 0.05 °×0.05 ° reanalysis data. The model performance slightly decreases when using the meteorological reanalysis compared to that driven by tower-derived meteorology data (please refer to the section 3.1 in the revised manuscript). To further mitigate the uncertainty, we used the parameters optimized by global reanalysis meteorology data to simulate the GPP at global scale.

And we discussed the uncertainty introduced by the mismatches between eddy covariance flux footprint and image pixels of the input dataset in section 4.3 Model uncertainty:

"Additionally, the uncertainty of the revised EC-LUE model may arise by scale mismatches between eddy covariance flux footprint and input dataset. The eddy covariance flux footprint is generally less than 3 $km^2$ and varies depending on the wind speed, wind direction and the atmospheric stability (Tan et al., 2006). In our studies, the revised EC-LUE model was run at 0.05 degree (~5 $km^2$) spatial resolution. The uncertainty of simulated GPP introduced by the scale effect is inevitable but smaller than that introduced by the model structures, parameters or input datasets (Sjostrom et al., 2013; Zheng et al., 2018)." (Line 392-396 in the revised manuscript)

**7. Lines 282-285. Needs more uncertainty analysis by factor (e.g., radiation) to support this.**

**Line 282-285: In contrast, 74% of the sites showed higher $R^2$ values (>0.5) for the revised EC-LUE model. The improvements of the revised EC-LUE model in reproducing interannual variations are owing to the integration of several important environmental drivers for vegetation production (i.e., atmospheric CO₂ concentration, radiation components, and VPD), which exhibited large variations and contributed significantly to vegetation production at interannual scale.**

Response: This statement is based on the results presented by Fig. 3 in the revised manuscript (namely Fig. 4 in the original manuscript). The comparison showed the revised EC-LUE model has the better performance for reproducing the interannual variability in GPP compared to the original EC-LUE and other LUE models. It is a very good idea to identify the contributions of various factors to improve the model ability. However, to our knowledge, there is no recognized methods to conduct the uncertainty analysis by factors, and it will be very interesting to develop this method. However, we appreciate your understanding that it will beyond the scope of this study, and this manuscript is data description paper and the main purpose is to introduce the model methods and describe the global dataset of GPP with long-term series.

[Figure]

Figure 3: Site percentage of (a) correlation coefficients ($R^2$), and (b) regression slopes between the model simulated and tower-based interannual variabilities in GPP. (c) Averaged values (error bars represent the standard deviations) of $R^2$ and slope for various LUE models. rEC-LUE$_{(T)}$ and rEC-LUE$_{(R)}$ indicate the revised EC-LUE models derived from tower-derived meteorology data and meteorological reanalysis dataset. The $R^2$ and slopes of the other seven LUE models (i.e., EC-LUE, VPRM, VPM, MODIS, CFlux, CFix, and CASA) in the figure were obtained from the study by Yuan et al. (2014). ** and * indicate a significant difference in statistic variables ($R^2$ and slope) between the rEC-LUE$_{(T)}$ and other LUE models (i.e., rEC-LUE$_{(T)}$ and other seven LUE models ) at p-value < 0.01 and p-value<0.05, respectively.

**8. Line 325. throughout the seasons? or different results by season? Again, seasonal analysis should be conducted.**
**Line 325: The revised EC-LUE model showed the lowest accuracy for the evergreen broadleaf forests in the tropic areas (Figs. 2–3).**
Response: "throughout the seasons". As the response of comment #4, we test the seasonal performance of the revised EC-LUE model for each site. We also added another index (Kendall's coefficient of rank correlation τ) to further quantify the agreement between the simulated and tower estimated GPP at seasonal patterns in the revised manuscript (Fig. 2d in the revised manuscript).

**9. Figure 8. Comparison by region (or continent) would make the paper robust. Are there any merits of using the proposed model in terms of the spatial domain?**
Response: As your suggestion, we added the comparison between our model and other models across bioclimatic zones in the Köppen-Geiger climate classification map (Beck et al., 2018) before the Fig. 8 (in the original manuscript). Because we have rearranged the figures in our manuscript, the comparison across bioclimatic zones is Fig. 7 in the revised manuscript. We added the following comparison:

"At regional scale, we compared the annual mean GPP between the revised EC-LUE model and other models across the bioclimatic zones in the Köppen-Geiger climate classification map (Beck et al., 2018) (Fig. 7). The GPP of the revised EC-LUE model was comparable to the mean value of other models for each bioclimatic zone (Fig. 7a). The GPP of different models exhibited large discrepancies in tropical regions (Af/Am/Aw) (Fig. 7a). The correlations ($R^2$) of GPP across all the bioclimatic zones between the revised EC-LUE model and other models ranged from 0.73 (LPX-Bern) to 0.95 (FLUXCOM

MARS, FLUXCOM RF) (Fig. 7b)." (Line 325-331 in the revised manuscript)

[Figure]

Figure 7: Comparisons of long-term (1982 to 2010s) averaged GPP between the revised EC-LUE model and other models across bioclimatic zones in the K öppen-Geiger climate classification map (Beck et al., 2018). (a) the regional averaged value (b) correlation coefficients ($R^2$) of GPP at all the bioclimatic zones between the revised EC-LUE model and other models. These models including machine learning models (FLUXCOM ANN, FLUXCOM MARS, FLUXCOM RF; Jung et al., 2017), biophysical models BEPS (Ju et al., 2006; Liu et al., 2018), and ten biophysical models in TRENDY (CABLE, CLASS, CLM, ISAM, JSBACH, JULES, LPJ-GUESS, LPX-Bern, ORCHIDEE, and VISIT). The abbreviations for the bioclimatic zones are as follows: Af, tropical, rainforest; Am, tropical, monsoon; Aw, tropical, savannah; BWh, arid, desert, hot; BWk, arid, desert, cold; BSh, arid, steppe, hot; BSk, arid, steppe, cold; Csa, temperate, dry summer, hot summer; Csb, temperate, dry summer, warm summer; Csc, temperate, dry summer, cold summer; Cwa, temperate, dry winter, hot summer; Cwb, temperate, dry winter, warm summer; Cwc, temperate, dry winter, cold summer; Cfa, temperate, no dry season, hot summer; Cfb temperate, no dry season, warm summer; Cfc, temperate, no dry season, cold summer; Dsa, cold, dry summer, hot summer; Dsb, cold, dry summer, warm summer; Dsc, cold, dry summer, cold summer; Dsd, cold, dry summer, very cold winter; Dwa, cold, dry winter, hot summer; Dwb, cold, dry winter, warm summer; Dwc, cold, dry winter, cold summer; Dwd, cold, dry winter, very cold winter; Dfa, cold, no dry season, hot summer; Dfb, cold, no dry season, warm summer; Dfc, cold, no dry season, cold summer; Dfd, cold, no dry season, very cold winter; ET, polar, tundra; EF, polar, frost.

**10. Lines 371-372. Don't see any conclusive results to say that the model has a unique superiority in reproducing the inter annual variations in GPP at both site level and global scales. Superiority to what? Any comparison with other models (e.g., machine learning or physical models) to show**

**the inter annual variations?**

**Line 371-372: The revised EC-LUE performed well in simulating the spatial, seasonal, and interannual variations in global GPP. Particularly, it has a unique superiority in reproducing the interannual variations in GPP at both site level and global scales.**

Response: In our manuscript, we compared the model performance at interannual variations of the revised EC-LUE model with other LUE models, such as the original EC-LUE model, CASA, CFix, CFlux, MODIS, VPM, and VPRM. The result showed the revised EC-LUE indeed has a unique superiority in reproducing interannual variations than other LUE models. Over the sites with longer 5-year observations, the averaged $R^2$ between annual mean tower-estimated and model simulated GPP are 0.44 for the revised EC-LUE model, which is significantly higher than those of original EC-LUE model ($R^2 = 0.36$) and other LUE models ($R^2$ ranged from 0.06 to 0.30 with an average value of 0.16), and these results have been represented at Fig. 3 (in the revised manuscript).

We appreciate your understanding that we don't compare it with other kind models because the main objective of our manuscript is focused on the improvement of the LUE models and produced a long-term GPP dataset. Moreover, due to large data gaps of measurements derived from eddy covariance towers, we need run process-based models at eddy covariance towers and obtain the corresponding simulations with observations of GPP in order to evaluate the model performance. This work needs contributions of model PIs, and which probably need take long time and great efforts. In addition, previous studies have provided the insights on this issue. Keenan et al. (2012) compared the performance of 16 process-based biophysical models and 3 satellite-based models (including the MODIS product) in reproducing the interannual variations in GPP. The result indicated the MODIS model performance was comparable to the process-based biophysical models. In our manuscript, the revised EC-LUE model (averaged $R^2 = 0.44$) was significantly better than the MODIS model (averaged $R^2 = 0.17$) at interannual scale. Therefore, we can conclude the similar result that our model was better than the process-based biophysical models compared in Keenan et al. (2012).

In order to emphasize that we conducted the comparison with other LUE models, we revised these sentences you mentioned (Line 371-372 in the original manuscript) as following:

"The revised EC-LUE performed well in simulating the spatial, seasonal, and interannual variations in GPP across the globe. Particularly, it has a unique superiority in reproducing the interannual variations in GPP ($R^2 = 0.44$) compared with the original EC-LUE model ($R^2 = 0.36$) and other LUE models ($R^2$ ranged from 0.06 to 0.30 with an average value of 0.16)." (Line 405-407 in the revised manuscript)

The comparisons with other LUE models are shown in the abstract, result, and discussion section.

In the abstract section:

[revised manuscript text omitted]

---

## Referee Report (RR1)

The paper entitled by 'Improved estimate of global gross primary production for reproducing its long-term variation, 1982-2017' proposed a new light use efficiency model (rEC-LUE) by integrating part of the Faquhar's model (a process model for leaf-scale photosynthesis of C3 plant) into EC-LUE model, which showed an obvious improvement of the simulation performance. Based on the simulations of this new model, the authors produced a new global GPP data very correlated to FLUXCOM GPP on annual scale. According to some of the Reviewer #2's comments, the authors also have provided adequate figures and revisions. However, I have the same opinion as the second one of Reviewer #2, which the authors didn't accept:

**In addition, it is necessary to closely look into a few specific sites (through selection of flux sites considering climate and other environmental factors) using time-series data. I guess the proposed model would work well for some sites, and not for others. Additional discussion should follow regarding that. When simply looking at Figure 8, the proposed model appears to somewhat underestimate GPP compared to the existing ones, which implies more validations are required on spatially and temporally detailed domains.**

It's very necessary that the authors could provide the time series result for some specific sites (can be one low-r2 site, one median -r2 site and one high-r2 site) and give some analysis about the bias between models against observations.

Furthermore, I think the authors should also address following questions in their paper:

To build this model:

1. The authors were using part of Farquhar's model (Equation 5) to represent the CO2 effect on GPP and embedding it into the EC-LUE model directly. However, this part of Farquhar's model only includes the limitation from the electron-transport efficiency (or Jmax) which is a super much simplification. The explanation should be given on why the limitation from Rubisco carboxylation rate (or Vcmax) and non-linear effect of Jmax which also include the CO2 effect were ignored.
2. In the introduction, the authors only talked about on the environmental factors they considered in their model, but didn't look into the other factors, for example: soil moisture, which is considered much more important than VPD in many other light use efficiency models (e.g. VPM, CASA, Horn's model and even the original EC-LUE model). The state of art on choosing the environmental factors should be investigated.

To force the models:

1. Because of weather impact, the EC sites also produce very bad quality data, which could not represent the CO2 exchange under real condition. To some extent, the bad quality data could be filtered out by the quality flag. To calibrate the parameters, only the good quality data should be used for all these models.
2. It's better to keep the observed GPP and simulated GPP on the same time scale when calibrating the parameters.

To evaluate the models:

1. Besides giving some examples for specific sites, the site-specific R2, RMSE and bias for each model should also be given either in a plot (a scatter, histogram or boxplot) or in a table (the

R2/RMSE/bias of the best model and rEC-LUE model for each site). The statistical hypothesis test between different models could help to find the best model.

2. Figure 3: the highest slope is still under 0.5, does it mean the GPP is underestimated for most sites?
3. Figure 4: most of the GPP are close to 0. Please check the data quality, if data is good, could try data only in growing season.
4. Could you give a reason why the r2 of FLUXCOM GPP is lower than their paper (Jung et al., 2017)?
5. Figure 6: it seems the negative GPP trend happen in the region where the uncertainty is relatively big. This should be discussed.
6. Figure 8: it's incredible that the GPP sensitivity to VPD is so much higher than to air temperature. Can you compare this with the sensitivity derived from other LUE model or GPP products?
7. Some letters in this paper look weird: 'v', 'x', 'z' and '%' are extremely small comparing other letters, please check the format before upload.

---

## Referee Report (RR2)

The authors have answered the questions adequately. The revised EC-LUE model could work well at site level and for the global grids (the R2 of 95% sites were larger than 0.5 and the gridded GPP had higher R2 and lower RMSE than machine learning models and TRENDY products). Although the models still underestimated annual GPP significantly (slope<0.5 in Figure 3 and smaller daily GPP in Figure S2-S3), it fitted the tendencies well (Figure S1-S3) and performed better than the other models in this paper. It highlighted the usefulness to integrate $CO_2$ in the LUE models to simulate GPP. I think it's worth to accept this paper after following minor revision:

1. Could you add the site R2/RMSE/Bias/tau of other models in figure 2 (can be the best one)?

2. The mean R2 of eight LUE models was mentioned but not determined in 3.1 and Figure S4.

---

## Author Response (AR2)

**Journal:** ESSD

**Title:** Improved estimate of global gross primary production for reproducing its long-term variation, 1982–2017

**MS No.:** essd-2019-126

**MS Type:** Data description paper

Dear Prof. Yuyu Zhou,

We appreciate the valuable opportunity to further revise our manuscript "Improved estimate of global gross primary production for reproducing its long-term variation, 1982–2017" (**MS No.:** essd-2019-126) for possible publication in ESSD.

The remaining issues pointed out by Reviewer#2 have been carefully addressed. Especially, the accuracy of the generated long-term GPP dataset and the robustness of the revised EC-LUE model have been further proved by adding more detailed validations, model comparisons and uncertainty analysis, as suggested by the reviewer. Two more figures have been added accordingly (Figs. 4 and 7); and, the corresponding texts in the Methods, Results, and Discussion sections have been carefully revised.

Please find attached the point-by-point responses to the comments of reviewer#2. Please note that the comments from the reviewers are in **bold** followed by our responses in regular text. The changes in our manuscript are underlined with red.

We believe the quality of the manuscript can now meet the high standard of ESSD and deeply appreciate your consideration of our manuscript.

Sincerely,

Yi Zheng, Wenping Yuan

School of Atmospheric Sciences,

Sun Yat-sen University, Zhuhai 519082, Guangdong, China

Email: yuanwpcn@126.com

**Response to Reviewer #2:**

**1. The manuscript has been improved according to the review comments raised by reviewers. However, I am still much concerned about the use of the proposed approach and the resultant data. Most of all, it is not convincing that the proposed EC-LUE model outperforms or is comparable to the existing good models.**

**For this type of research requires a series validations to support the robustness of the proposed model. This study did some validations, but all of them were smoothed through spatio-temporal aggregation. For example, Figure 1 shows the annually aggregated data from the proposed model. Temporally detailed plots (density plots using monthly data) would be better to support the robustness of the model. In addition, a few other well-defined models can be used to generated such scatterplots for comparison. It is not clear whether the proposed model produces accurate GPP data or not. Considering the plot based GPP data as reference ones, it is crucial to compare the proposed model-derived GPP to other physical models' output in terms of the reference data, which is not shown in this paper.**

We understand your concern. With respect to the concern on the mixed spatial and temporal validations, we have previously conducted the model validations separately at spatial and temporal scales. Fig. 1 shows the validation results at the spatial scale and indicate a good reproduction of the spatial GPP variations by our model. Temporally detailed plots have been shown as Fig. 2 to examine if the model can accurately reproduce the temporal variations of 8-day GPP at each site, as indicated by the indices of $R^2$, RMSE, bias, and seasonal index ($\tau$).

Besides, thanks to the great suggestion by the reviewer on model comparisons, the revised EC-LUE model has been proved to be superior to other models. In the revised manuscript, we have included other seven light use efficiency models, three machine learning models and ten process-based ecosystem models, and compared our model with these models using scatterplots at monthly step against the GPP based on eddy covariance measurements, as suggested by the reviewer. The Methods and Results sections have been modified accordingly as follows:

In the Methods section:

"In addition, we compared the model performance of the revised EC-LUE model with seven light use efficiency models, three machine learning methods and ten process-based ecosystem models. The participated light use efficiency models include CASA (Potter et al., 1993), CFlux (Turner et al., 2006; King et al., 2011), CFix (Veroustraete et al., 2002), MODIS (Running et al., 2004), VPM (Xiao et al., 2005), VPRM (Mahadevan et al., 2008), and EC-LUE (Yuan et al., 2007). We calibrated the model parameters of all seven light use efficiency models based on the eddy covariance measurements using the same parameterization method as the revised EC-LUE model (see the above method), and then compared the GPP simulations of seven LUE models driven by EC tower-based meteorology data against the estimated GPP based on EC measurements. For the comparison with machine learning models and process-based ecosystem models, we collected their global GPP products released by FLUXCOM (Jung et al., 2017) and TRENDY program (version 5) (Le Quéré et al., 2016), respectively. FLUXCOM program uses the Artificial Neural Network method (FLUXCOM ANN), the Multivariate Adaptive Regression Splines method (FLUXCOM MARS), and the Random Forest method (FLUXCOM RF); and TRENDY program includes the CSIRO Atmosphere and Biosphere Land Exchange (CABLE) (Zhang et al., 2013), the coupled Canadian Land Surface Scheme and Canadian Terrestrial Ecosystem Model (CLASS-CTEM) (Melton and Arora, 2016), the Community Land Model (CLM) (Oleson et al., 2013), the Integrated Science Assessment Model (ISAM) (Jain et al., 2013), the land component of the Max Planck Institute Earth System Model (JSBACH) (Reick et al., 2013), the Joint UK Land Environment Simulator (JULES) (Clark et al., 2011), the Lund-Postdam-Jena General Ecosystem Simulator (LPJ-GUESS) (Smith et al., 2014), the Land surface Processes and eXchanges (LPX-Bern) (Stocker et al., 2014), the ORganizing Carbon and Hydrology In Dynamic EcosystEms (ORCHIDEE) (Krinner et al., 2005), and the Vegetation Integrated Simulator for Trace Gases (VISIT) (Kato et al., 2013). The monthly GPP simulations at all investigated EC sites were derived from their global products, and equally we obtained the monthly GPP simulations of the revised EC-LUE model from its global dataset driven by MERRA-2 reanalysis dataset." (Lines 211-231 in the revised manuscript)

In the Results section:

"Additionally, we examined the model performance of the revised EC-LUE model at monthly step by comparing with other LUE models, machine learning models, and processed-based models in TRENDY (Fig. 4). In comparison with seven LUE models driven by EC tower-based meteorology dataset, the $R^2$ of the revised EC-LUE model was 0.76, higher than the original EC-LUE model and other LUE models ($R^2$ ranged from 0.65 to 0.71) (Fig. 4a). By using the global reanalysis

meteorology data, we compared the performance of the revised EC-LUE model with three existing machine learning model products and ten process-based model products in TRENDY (Fig. 4b). The $R^2$ of the revised EC-LUE model ($R^2 = 0.66$) was higher than that of other models ($R^2$ ranged from 0.06 to 0.61) (Fig. 4b)." (Lines 282-288 in the revised manuscript)

[Figure]

Figure 4: Comparisons between estimated GPP based on EC measurements and GPP simulations by the various models (including LUE models, machine learning models, and process-based models in TRENDY) at monthly scale. The comparison of GPP simulations were simulated using (a) tower-derived meteorology data and (b) global reanalysis meteorology data, respectively (see method 2.4). rEC-LUE$_{(T)}$ and rEC-LUE$_{(R)}$ indicate the simulations of the revised EC-LUE model derived from tower-derived meteorology data and global reanalysis meteorology data, respectively.

**2. In addition, it is necessary to closely look into a few specific sites (through selection of flux sites considering climate and other environmental factors) using time-series data. I guess the proposed model would work well for some sites, and not for others. Additional discussion should follow regarding that. When simply looking at Figure 8, the proposed model appears to somewhat underestimate GPP compared to the existing ones, which implies more validations are required on spatially and temporally detailed domains.**

It is a good idea to examine the model by using time-series data at a few specific sites. This study has examined the model performance over each of the examined 95 sites (see the Fig. 2), and we provided the statistical results for all sites in order that the readers can know the performance at all sites. We agreed with the reviewer that none of models can work well for all sites, including the revised EC-LUE model. In the previous version, there is one paragraph to introduce the differences of model performance among different vegetation types.

"The revised EC-LUE model also showed a good performance in reproducing the seasonal variations in GPP at most EC sites (Fig. 2). In this study, we compared the modeled and tower GPP at 8-day step for each site to examine the model capacity in reproducing the temporal variations of GPP. In terms of GPP simulations driven by tower-derived meteorology data, the coefficients of determination ($R^2$) varied from 0.26 at MY-PSO site to 0.96 at DK-Sor site, with most of them being statistically significant (*p*-value <0.05) (Fig. 2a), and the mean $R^2$ was 0.81 over all investigated sites. The low $R^2$ values (<0.4) were found at three tropical forest sites (i.e., MY-PSO, BR-Sa1 and BR-Sa3). The averaged Kendall's correlation coefficient ($\tau$) was 0.63 over all sites, indicating a strong seasonal coherence between simulated and tower-estimated GPP (Fig. 2d). Similarly, $\tau$ at tropical forest sites were generally lower than other sites. According to the RMSE and absolute bias, the revised EC-LUE model performed very well at most sites. The averaged RMSE and absolute bias over all the sites were 2.13 and 0.81 g C m$^{-2}$ d$^{-1}$, respectively (Fig. 2b–c). In addition, there was no obvious difference between the seasonal GPP performances using the tower-derived meteorology data and the meteorological reanalysis dataset (Fig. 2)." (Lines 258-268 in the revised manuscript)

We also discussed the reasons on the low performance at some sites in section 4.3:

"Similarly, other satellite-based models exhibited a large uncertainty in the GPP simulations over tropical forest areas (Ryu et al., 2011; Yuan et al., 2014). For example, MODIS GPP product (MOD17) underestimated GPP at high productivity sites over the tropical evergreen forests (de Almeida et al., 2018). Regarding the quality of satellite data, a high cloud cover exists over tropical regions, introducing large uncertainties to FAPAR/LAI and other vegetation indices (e.g., NDVI and EVI). As suggested by de Almeida et al. (2018), the lack of reliable MOD15 FAPAR data during January to April as a result of the cloudiness contamination could have substantially affected the seasonality of GPP estimates. Besides, the quality of satellite data can even affect the evaluation of the interannual variations in GPP. Using MODIS EVI data, Saleska et al. (2007) reported a large-scale green up in the Amazon evergreen forests during the drought of 2005. However, an opposite conclusion was drawn when the cloud-contaminated data were excluded from the analysis (Samanta et al., 2010). Subsequent studies noted an increase in LAI and EVI during the dry season in the Amazon evergreen forests; however, a recent study highlighted that the apparent seasonal changes in EVI were resulted from the variations in the sun-sensor geometry rather than vegetation greenness (Morton et al., 2014)." (Lines 389-400 in the revised manuscript)

The reviewer also concerns that the model may underestimate the global GPP compared to other models according to Fig. 8. It should be noticed that Fig. 8 (now Fig. 10 in the revised manuscript) shows the comparison of global GPP magnitude among various models. It is the best way for judging the model performance to compare the simulated GPP against observations as shown in Fig. 1, Fig. 2, and Fig. 4 (please refer the response #1). A superior performance is indicated in the revised EC-LUE model as compared to the existing ones.

**3. Regarding the uncertainty analysis, it would make the paper much robust if the authors provided uncertainty maps, not just narrative discussion based on potential factors that may result in uncertainty. More quantitative analysis should follow.**

Thanks for this insightful suggestion. We have added the uncertainty map of the simulated GPP as Fig. 7 in the revised manuscript and modified the Methods, Results, and Discussion sections accordingly as follows:

In the Methods section:

"In order to investigate the uncertainties of the global GPP dataset, 10,000 sets of optimized parameters were randomly selected to simulate global GPP by assuming a normal distribution of these parameters (Table 3). The uncertainty of global GPP simulations was determined by the mean absolute deviation (MAD) of all the 10,000 simulations (Khair et al., 2017)."

(Lines 195-198 in the revised manuscript)

In the Results section:

"In addition, this study used the MAD of 10,000 simulations to quantify the uncertainty of estimated GPP globally (see methods). Over the globe, the mean uncertainty of estimated GPP by the revised EC-LUE model is 19.33 Pg C yr$^{-1}$. The GPP uncertainties were found to be low in high and middle latitudes, but relatively high in tropical forests (about 600 g C m$^{-2}$ yr$^{-1}$) (Fig. 7)." (Lines 305-307 in the revised manuscript)

In the Discussion section:

"The uncertainties of our GPP dataset were low in high and middle latitude areas but high in tropical areas (Fig. 7). This is consistent with the validations at site level that the revised EC-LUE model showed the lowest accuracy over the tropical evergreen broadleaf forest sites (Fig. 2)." (Lines 387-389 in the revised manuscript)

[Figure]

Figure 7: Spatial pattern of the uncertainty in global GPP simulated by the revised EC-LUE model.

**4. Overall, I agree that the authors put enormous effort on revising the model and comparing the proposed model-based GPP with other global models' output. However, I feel more (spatio-temporally) detailed validations and comparisons are necessary to evaluate the proposed model. ESSD is a data journal, which means if the accuracy of data is not well supported, they would not appeal the users.**

We appreciate the reviewer's constructive comments and great helps in further improving the manuscript. The remaining issues pointed out by the reviewer have been carefully addressed by 1) adding Fig. 4 to compare the performance of the revised EC-LUE model with other seven light use efficiency models, three machine learning models and ten process-based ecosystem models, and 2) adding Fig. 5 to show the uncertainty map of the simulated global GPP determined by the mean absolute deviation of 10,000 sets of optimized parameters. These additional validations and comparisons as suggested by the reviewer have helped in further proving that the revised EC-LUE model is a robust model for generating long-term GPP dataset globally.

**References**

[revised manuscript text omitted]

---

## Author Response (AR3)

**Journal:** ESSD

**Title:** Improved estimate of global gross primary production for reproducing its long-term variation, 1982–2017

**MS No.:** essd-2019-126

**MS Type:** Data description paper

Dear Prof. Yuyu Zhou,

We appreciate the valuable opportunity to further revise our manuscript "Improved estimate of global gross primary production for reproducing its long-term variation, 1982–2017" (**MS No.:** essd-2019-126) for possible publication in ESSD.

We have carefully studied these comments and substantially revised our manuscript accordingly. Please find attached the point-by-point responses to the comments of the reviewer. Please note that the comments from the reviewers are in **bold** followed by our responses in regular text. The changes in our manuscript are underlined with red.

We believe the quality of the manuscript can now meet the high standard of ESSD and deeply appreciate your consideration of our manuscript.

Sincerely,

Yi Zheng, Wenping Yuan

School of Atmospheric Sciences,

Sun Yat-sen University, Zhuhai 519082, Guangdong, China

Email: yuanwpcn@126.com

**Response to Reviewer:**

The paper entitled by 'Improved estimate of global gross primary production for reproducing its longterm variation, 1982-2017' proposed a new light use efficiency model (rEC-LUE) by integrating part of the Faquhar's model (a process model for leaf-scale photosynthesis of C3 plant) into EC-LUE model, which showed an obvious improvement of the simulation performance. Based on the simulations of this new model, the authors produced a new global GPP data very correlated to FLUXCOM GPP on annual scale. According to some of the Reviewer #2's comments, the authors also have provided adequate figures and revisions. However, I have the same opinion as the second one of Reviewer #2, which the authors didn't accept:

In addition, it is necessary to closely look into a few specific sites (through selection of flux sites considering climate and other environmental factors) using time-series data. I guess the proposed model would work well for some sites, and not for others. Additional discussion should follow regarding that. When simply looking at Figure 8, the proposed model appears to somewhat underestimate GPP compared to the existing ones, which implies more validations are required on spatially and temporally detailed domains.

It's very necessary that the authors could provide the time series result for some specific sites (can be one low-$r^2$ site, one median -$r^2$ site and one high-$r^2$ site) and give some analysis about the bias between models against observations.

Thanks for your comments. In our study, the model performed well at most sites except several tropical forest sites (i.e., MY-PSO, BR-Sa1 and BR-Sa3) without pronounced seasonal variations (Fig. 2). Therefore, we selected three sites with low $R^2$ (Br-Sa3), median $R^2$ (CN-Din), and high $R^2$ (US-UMB) to illustrate the time-series changes of observed/simulated GPP and environmental factors (Figs S1-S3). The new analyses have been added into the revised manuscript:

"Furthermore, we selected three sites with high $R^2$ (US-UMB; DBF; $R^2 = 0.93$), median $R^2$ (CN-Din; EBF; $R^2 = 0.71$), and low $R^2$ (Br-Sa3; EBF; $R^2 = 0.39$) to illustrate the time-series changes of observed/simulated GPP, LAI, and environmental factors (i.e., air temperature, VPD, and PAR) (Figs S1-S3). At US-UMB site, the model captured the GPP variations well all the year round with no obvious bias (Fig. S1). At CN-Din site, the model generally performed well except the underestimation at the end of the year (November-December) with decreased LAI. While, at Br-Sa3 site, the model could not capture the variations of GPP for the vegetation greenness and environmental factors varying slightly during the year (Fig. S3)." (Lines 271-277 in the revised manuscript)

[Figure]

**Figure S1: Time-series changes of (a) tower estimated GPP, model simulated GPP, and LAI, and (b) environmental factors (i.e., air temperature, VPD, and PAR) for low R$^2$ site in Fig. 2, taking US-UMB site as an example.**

[Figure]

**Figure S2: Time-series changes of (a) tower estimated GPP, model simulated GPP, and LAI, and (b) environmental factors (i.e., air temperature, VPD, and PAR) for low $R^2$ site in Fig. 2, taking CN-Din site as an example.**

[Figure]

**Figure S3: Time-series changes of (a) tower estimated GPP, model simulated GPP, and LAI, and (b) environmental factors (i.e., air temperature, VPD, and PAR) for low R$^2$ site in Fig. 2, taking Br-Sa3 site as an example.**

**Furthermore, I think the authors should also address following questions in their paper:**

**To build this model:**

**1. The authors were using part of Farquhar's model (Equation 5) to represent the CO$_2$ effect on GPP and embedding it into the EC-LUE model directly. However, this part of Farquhar's model only includes the limitation from the electron-transport efficiency (or Jmax) which is a super much simplification. The explanation should be given on why the limitation from Rubisco carboxylation rate (or Vcmax) and non-linear effect of Jmax which also include the CO$_2$ effect were ignored.**

$$C_s = \frac{C_i - \varphi}{C_i + 2\varphi} \tag{5}$$

In this study, we considered the atmosphere CO$_2$ fertilization effect on vegetation production, and used the equations (i.e. Eqs. 5-11) of Farquhar's model to simulate leaf internal CO$_2$ concentration. These equations are not for simulating photosynthesis rate limited by electron-transport efficiency (i.e. Jmax). As you know, EC-LUE model is based on the principle of light use efficiency model, which is different to Farquhar's model.

**2. In the introduction, the authors only talked about on the environmental factors they considered in their model, but didn't look into the other factors, for example: soil moisture, which is considered much more important than VPD in many other light use efficiency models (e.g. VPM, CASA, Horn's model and even the original EC-LUE model). The state of art on choosing the environmental factors should be investigated.**

Thanks for your deep thoughts. Indeed, soil moisture is a very important factor to impact vegetation production. Although, there are a few satellite-based soil moisture products available currently, such as SMAP, and AMSR-E soil moisture products. However, there are large data gaps across the spatial and temporal scales. In addition, many process-based terrestrial ecosystem models represent the effects of reduced soil moisture on canopy carbon assimilation using an empirical drought factor commonly referred as b factor (Cox et al., 1998). The b-factor approach has been shown to overestimate plant responses to seasonal and experimentally induced drought (Ukkola et al., 2016; Eller et al., 2018). The b-factor has a large impact on the modelled global carbon budget, supressing 30–40% of the annual gross primary productivity (GPP) in large areas of arid and semiarid ecosystems (Trugman et al., 2018). Therefore, none of light use efficiency models used satellite-based soil moisture to indicate the water stress of vegetation production. VPM model used satellite-based normalized difference water index (NDWI; Xiao et al., 2005) to indicate moisture condition of vegetation canopy. CASA and original EC-LUE model used the ratio of ecosystem evapotranspiration to potential evapotranspiration or net radiation (Potter et al., 1993; Yuan et al., 2007).

In the revised EC-LUE model, we used VPD as the water scalar to simulate the impacts of water stress on vegetation production based on the following two reasons. First, VPD plays an important role in regulating photosynthesis. Previous numerous studies have highlighted that the leaf and canopy photosynthetic rate declines when the atmospheric VPD increases due to stomatal closure (Fletcher et al., 2007; Grossiord et al., 2020). A recent study highlighted that increases in VPD rather than changes in precipitation would be a dominant influence on vegetation productivity (Konings et al., 2017). Second, there is a substantial long-term change of VPD driven by climate warming. Rising air temperature increases the saturated vapor pressure at a rate of ~7%/K according to the Classius–Clapeyron relationship, and therefore, VPD will increase if the atmospheric water vapor content does not increase by exactly the same amount as the saturated vapor pressure. Numerous studies indicated significant changes in the VPD in both humid areas and continental areas located far from oceanic humidity (Van Wijngaarden and Vincent, 2004; Simmons et al. 2010; Pierce et al., 2013; Willett et al. 2014). Our recent study also shows the strong regulations of long-term VPD changes to global vegetation growth (Yuan et al., 2019), and thus it is necessary to integrate the VPD impacts into the light use efficiency models. In addition, we did not take more efforts to integrate the impacts of soil moisture on vegetation production, mostly because there are strong correlations of interannual variability between VPD and soil moisture. Several previous studies have reported highly consistent interannual variability of VPD and soil moisture (Zhou et al., 2019a, b).

In our manuscript, there is one paragraph to discuss the water stress in our model:

"The evaporation fraction (EF), namely the ratio of evapotranspiration (ET) to net radiation (Rn), was used to indicate the water stress on vegetation growth in the original EC-LUE model (Yuan et al., 2007; 2010). While the atmospheric VPD was used to indicate water stress to avoid the aggregated errors from ET simulations in the revised EC-LUE model. Physiologically, vegetation production is sensitive to both atmospheric VPD and soil moisture availability to roots. Several studies have reported highly consistent interannual variability of VPD and soil moisture (Zhou et al., 2019a, b). In addition,

recent studies highlighted that the increase in VPD had a larger limitation to the surface conductance and evapotranspiration than soil moisture over short time scales in many biomes (Novick et al., 2016; Sulman et al., 2016). Other studies have also suggested substantial impacts of VPD on vegetation growth (de Cárcer et al., 2018; Ding et al., 2018), forest mortality (Williams et al., 2013), and crop yields (Lobell et al., 2014). It is increasingly important to integrate the atmospheric water constraint to the carbon and water flux modeling." (Lines 364-373 in the revised manuscript)

**To force the models:**

**1. Because of weather impact, the EC sites also produce very bad quality data, which could not represent the $CO_2$ exchange under real condition. To some extent, the bad quality data could be filtered out by the quality flag. To calibrate the parameters, only the good quality data should be used for all these models.**

In our manuscript, we used the quality flags of each variable to exclude the bad quality data with less than 20% of good data. We revised manuscript as follow:

"In the FLUXNET dataset, there were quality flag ranged from 0 to 1 to indicate percentage of measured and good quality gap-filled data. For each variable, we used the daily/monthly values with more than 80% of good quality data (quality flag > 0.8). We aggregated the daily values to 8-day time step. And only the 8-day measurements with more than 5-day valid values were used." (Lines 121-124 in the revised manuscript)

**2. It's better to keep the observed GPP and simulated GPP on the same time scale when calibrating the parameters.**

Yes, we calibrated the parameters using flux tower GPP and other variables at 8-day time scale, and we validated the model (Figs 1-2) and produced the GPP product at the same time scale (8-day). For the main objective of our manuscript is focused on the improvement of the LUE models in interannual variations and produced a long-term GPP dataset. We aggregated the 8-day simulated GPP to yearly scale to examine the ability of the LUE models in reproducing the interannual variations in GPP (Fig. 3).

**To evaluate the models:**

**1. Besides giving some examples for specific sites, the site-specific $R^2$, RMSE and bias for each model should also be given either in a plot (a scatter, histogram or boxplot) or in a table (the $R^2$/RMSE/bias of the best model and rEC-LUE model for each site). The statistical hypothesis test between different models could help to find the best model.**

In the revised manuscript, we evaluated the performance of each model in Fig.4 at each site by comparing the simulated GPP against the tower estimated GPP at monthly step using $R^2$, RMSE, and absolute value of bias. For each site, we compared the $R^2$/RMSE/absolute value of bias of the individual model with the averaged value of all the models (one site has an averaged $R^2$/RMSE/absolute value of bias). Then we added a boxplot to show which model in Fig. 4a and Fig. 4b had more sites with higher $R^2$/lower RMSE/lower absolute value of bias than the averaged values of all the models (Fig. S4). The Results section was modified accordingly as follows:

"Additionally, we examined the model performance of the revised EC-LUE model, other LUE models, machine learning models, and processed-based models in TRENDY at monthly step by comparing against EC tower estimated GPP (Fig. 4). In comparison with seven LUE models driven by EC tower-based meteorology dataset, the overall $R^2$ of the revised EC-LUE

model was 0.71, higher than the original EC-LUE model and other LUE models ($R^2$ ranged from 0.55 to 0.61) (Fig. 4a). For each site, we compared the $R^2$/RMSE/absolute value of bias of the individual model with the averaged value of all the eight LUE models (each site has an averaged $R^2$/RMSE/absolute value of bias) (Fig. S4a1-c1). The revised EC-LUE model had higher $R^2$ than the mean $R^2$ of the eight LUE models at 62% sites, which was comparable with the original EC-LUE model (63% sites) and VPM model (60% sites) (Fig. S4a1). Moreover, the revised EC-LUE model showed the lower RMSE and bias compared to mean values of all eight LUE models at 68% and 67% sites respectively, which indicated the better performance compared to the other LUE models at most sites (Fig. S4b1-c1). By using the global reanalysis meteorology data, we compared the performance of the revised EC-LUE model with three existing machine learning model products and ten process-based model products in TRENDY (Fig. 4b). The overall $R^2$ of the revised EC-LUE model ($R^2 = 0.57$) was higher than that of other models ($R^2$ ranged from 0.02 to 0.54) (Fig. 4b). The revised EC-LUE model, FLUXCOM ANN, and FLUXCOM MARS had more sites (over 90%) with higher $R^2$ than the mean $R^2$ (Fig. S4a2). And the revised EC-LUE model, FLUXCOM MARS, and FLUXCOM RF showed the lower RMSE at more than 90% sites (Fig. S4b2). Compared to the other models, the revised EC-LUE model had highest site percentage (81%) with lower absolute value of bias (Fig. S4c2)." (Lines 291-306 in the revised manuscript)

[Figure]

**Figure S4: Percentage of sites for each model in Fig.4 where (a1, a2) $R^2$ of individual model > averaged $R^2$ of all models, (b1, b2) RMSE of individual model < averaged RMSE of all models, and (c1, c2) RMSE of individual model < averaged**

**absolute value of bias of all models. All the monthly GPP simulations in (a1)-(c1) derived from tower-derived meteorology data, and all the monthly GPP simulations in (a2)-(c2) derived from global reanalysis meteorology data.**

Besides the model comparison at monthly step, we compared the overall accuracy of the revised EC-LUE model and the original EC-LUE model in Fig. 2 using four metrics ($R^2$/RMSE/bias/seasonal index $\tau$). The result indicated the performance of the revised EC-LUE model was better than the original EC-LUE model. Yuan et al. (2014) compared the performance of seven models (i.e., EC-LUE, VPRM, VPM, MODIS, CFlux, CFix, and CASA), and the EC-LUE model and CFlux performed better than other models. Therefore, we can conclude the result that the revised EC-LUE model was in high accuracy when compared with other EC-LUE models. We updated Fig. 2 and the results:

"On average, the revised EC-LUE model showed higher $R^2$ and $\tau$, and lower bias and RMSE than the original EC-LUE model for both (Fig. 2)." (Lines 270-271)

[Figure]

**Figure 2: Comparisons of 8-day mean GPP between the model simulated GPP and tower estimated GPP. Solid and open dots indicate the GPP simulations of the revised EC-LUE model derived from tower-derived meteorology data and meteorological reanalysis dataset, respectively; solid and open squares indicate the GPP simulations of the original EC-LUE model derived from tower-derived meteorology data and meteorological reanalysis dataset, respectively.**

**2. Figure 3: the highest slope is still under 0.5, does it mean the GPP is underestimated for most sites?**

[Figure]

Figure 3: Site percentage of (a) correlation coefficients ($R^2$), and (b) regression slopes between the model simulated and tower-based interannual variabilities in GPP. (c) Averaged values (error bars represent the standard deviations) of $R^2$ and slope for various LUE models. rEC-LUE$_{(T)}$ and rEC-LUE$_{(R)}$ indicate the revised EC-LUE models derived from tower-derived meteorology data and meteorological reanalysis dataset. The $R^2$ and slopes of the other seven LUE models (i.e., EC-LUE, VPRM, VPM, MODIS, CFlux, CFix, and CASA) in the figure were obtained from the study by Yuan et al. (2014). ** and * indicate a significant difference in statistic variables ($R^2$ and slope) between the rEC-LUE$_{(T)}$ and other LUE models (i.e., rEC-LUE$_{(T)}$ and other seven LUE models) at *p*-value < 0.01 and *p*-value<0.05, respectively.

Yes, it is indeed. Although the revised EC-LUE model has improved the performance for reproducing interannual variability of GPP, there are still uncertainties in simulating interannual variability of GPP. As we discussed in the manuscript, we did not integrate the regulation of soil nitrogen content on vegetation production. Atmospheric nitrogen deposition has exhibited a large increasing trend in the past few decades because of the excessive fossil fuel combustion in the industrial and transportation sectors and the abuse of nitrogenous fertilizer in the agricultural practice (Galloway et al., 2004). However, leaf water, starch, lignin, and cellulose overlap with the absorption characters of nitrogen in the shortwave infrared bands, making it difficult to retrieve the nitrogen content (Kokaly and Clark, 1999). What's more, canopy structures, background, and illumination/viewing geometry can further decrease the capacity to detect leaf nitrogen (Yoder and Pettigrew-Crosby, 1995; Knyazikhin et al., 2013). Therefore, the future studies need strengthen the algorithm development of nitrogen impacts on vegetation production, and we discuss this issue in the manuscript (Line 426-440 in the revised manuscript).

**3. Figure 4: most of the GPP are close to 0. Please check the data quality, if data is good, could try data only in growing season.**

Thanks for your constructive comment. Yes, we excluded the measurements with poor data quality using quality flag

(refer comment #1 in "To force the models") and non-growing season. For it is difficult to define growing season globally, we used 0°C as the temperature threshold of growing season. We updated the results:

"Additionally, we examined the model performance of the revised EC-LUE model, other LUE models, machine learning models, and processed-based models in TRENDY at monthly step by comparing against EC tower estimated GPP (Fig. 4). In comparison with seven LUE models driven by EC tower-based meteorology dataset, the overall $R^2$ of the revised EC-LUE model was 0.71, higher than the original EC-LUE model and other LUE models ($R^2$ ranged from 0.55 to 0.61) (Fig. 4a). For each site, we compared the $R^2$/RMSE/absolute value of bias of the individual model with the averaged value of all the eight LUE models (each site has an averaged $R^2$/RMSE/absolute value of bias) (Fig. S4a1-c1). The revised EC-LUE model had higher $R^2$ than the mean $R^2$ of the eight LUE models at 62% sites, which was comparable with the original EC-LUE model (63% sites) and VPM model (60% sites) (Fig. S4a1). Moreover, the revised EC-LUE model showed the lower RMSE and bias compared to mean values of all eight LUE models at 68% and 67% sites respectively, which indicated the better performance compared to the other LUE models at most sites (Fig. S4b1-c1). By using the global reanalysis meteorology data, we compared the performance of the revised EC-LUE model with three existing machine learning model products and ten process-based model products in TRENDY (Fig. 4b). The overall $R^2$ of the revised EC-LUE model ($R^2 = 0.57$) was higher than that of other models ($R^2$ ranged from 0.02 to 0.54) (Fig. 4b)." (Lines 291-303 in the revised manuscript)

[Figure]

**Figure 4: Comparisons between estimated GPP based on EC measurements and GPP simulations in growing season (defined as temperature larger than 0°C) by the various models (including LUE models, machine learning models, and process-based models in TRENDY) at monthly scale. The comparison of GPP simulations were simulated using (a) tower-derived meteorology data and (b) global reanalysis meteorology data, respectively (see method 2.4). rEC-LUE(T) and rEC-LUE(R) indicate the simulations of the revised EC-LUE model derived from tower-derived meteorology data and global reanalysis meteorology data, respectively.**

**4. Could you give a reason why the r² of FLUXCOM GPP is lower than their paper (Jung et al., 2017)?**

Jung et al. (2017) assessed the consistency of global GPP anomaly patterns between FLUXCOM and TRENDY datasets, and didn't validate the FLUXCOM GPP products against the tower-based GPP. The reviewer may talk about the paper of Tramontana et al. (2016), which developed four machine learning models (FLUXCOM ANN, MARS, RF, and KRR) and evaluated the performance of FLUXCOM products based on the measurements of eddy covariance towers.

Our validation result is different with that of Tramontana et al (2016), and there are three major causes as followings:

(1) Tramontana et al (2016) and this study used difference forcing data when conducted the model validation, which may be the most important cause for different validation results. Tramontana et al. (2016) developed and validated FLUXCOM ANN, MARS, RF, and KRR models based on the in situ measured meteorology data. On contrary, this study used global FLUXCOM products driven by the reanalysis meteorological datasets. The uncertainties of global reanalysis meteorological datasets may largely decrease the accuracy of global FLUXCOM products.

(2) Tramontana et al (2016) developed four machine learning models (i.e., RF, MARS, ANN, and KRR), and validated the accuracy based on the median estimates of four methods against the measurements of eddy covariance towers. However, there are only the products of three methods (ANN, MARS, and RF) available for global analyses. In our manuscript, we directly validated three products individually, and did not use median values of three products, which may be another cause for lower $R^2$ of FLUXCOME GPP. In addition, KRR product is not available, and thus we cannot evaluate the contribution of KRR methods to validation of Tramontana et al (2016).

(3) Tramontana et al. (2016) developed and evaluated the four machine learning models using the La Thuile dataset, while we validated the monthly FLUXCOM GPP products using the FLUXNET2015 dataset. The two datasets have differences in site coverage, study period, and data processing methods, which probably induce some differences.

**5. Figure 6: it seems the negative GPP trend happen in the region where the uncertainty is relatively big. This should be discussed.**

[Figure]

**Figure 6: Spatial pattern of global GPP trend simulated by the revised EC-LUE models during 1982–2017: (a) trend of annual GPP, (b) trend of annual GPP at different temperature and precipitation gradients.**

[Figure]

**Figure 7: Spatial pattern of the uncertainty in global GPP simulated by the revised EC-LUE model.**

The GPP uncertainties were high in tropical forests (Fig. 7). It may be introduced by cloudiness contamination and the quality of optical satellite data. Coincidentally, a decrease of GPP was found in these areas in our study. Similarly, many studies reported a decrease of tropic carbon by using satellite-based optical and microwave data (Samanta et al., 2010; Fan et al., 2019; Liu et al., 2015). We discussed these issues in our manuscript.

"The uncertainties of our GPP dataset were low in high and middle latitude areas but high in tropical areas (Fig. 7). This is consistent with the validations at site level that the revised EC-LUE model showed the lowest accuracy over the tropical evergreen broadleaf forest sites (Fig. 2). Similarly, other satellite-based models exhibited a large uncertainty in the GPP simulations over tropical forest areas (Ryu et al., 2011; Yuan et al., 2014). For example, MODIS GPP product (MOD17) underestimated GPP at high productivity sites over the tropical evergreen forests (de Almeida et al., 2018). Regarding the quality of satellite data, a high cloud cover exists over tropical regions, introducing large uncertainties to FAPAR/LAI and other vegetation indices (e.g., NDVI and EVI). As suggested by de Almeida et al. (2018), the lack of reliable MOD15 FAPAR data during January to April as a result of the cloudiness contamination could have substantially affected the seasonality of GPP estimates. Besides, the quality of satellite data can even affect the evaluation of the interannual variations in GPP. Using MODIS EVI data, Saleska et al. (2007) reported a large-scale green up in the Amazon evergreen forests during the drought of 2005. However, an opposite conclusion was drawn when the cloud-contaminated data were excluded from the analysis (Samanta et al., 2010). In our study, a significant decrease of GPP was found in the Amazon evergreen forests, which may be resulted from the sharp increase in VPD after the late 1990s (Yuan et al., 2019). Studies using optical satellite data can be influenced by the cloudiness contamination. Recently studies using cloud free satellite-based microwave data also reported a carbon loss in tropic forest (Liu et al., 2015; Fan et al., 2019)." (Lines 406-420 in the revised manuscript)

**6. Figure 8: it's incredible that the GPP sensitivity to VPD is so much higher than to air temperature. Can you compare this with the sensitivity derived from other LUE model or GPP products?**

[Figure]

**Figure 8: Long-term changes in global GPP and the environmental regulations: (a) Global summed GPP derived from the four experimental simulations in section 2.5, (b) GPP sensitivity to climate variables (i.e., VPD, Ta, and PAR), LAI, and atmospheric $CO_2$, (c) contributions of climate variables (i.e., VPD, Ta, and PAR), LAI, and atmospheric $CO_2$ to GPP changes over 1982–2017, 1982–2000, and 2001–2017. * indicates the significant level at *p*-value<0.05.**

In our manuscript, global GPP decreased by 6.67 ±5.04 Pg C with an increased VPD of 0.1 kPa and global GPP slightly decreased by about 0.5 Pg C with an increased temperature of 1℃. GPP sensitivity to VPD (0.1 kPa) was so much higher than to air temperature (1℃). Increased temperature has a positive effect on vegetation productivity at high latitude and a negative effect on vegetation productivity at low latitude. The influence of increased temperature on vegetation growth will partly counteract globally. While the increased VPD suppresses vegetation growth at both low and high latitude. Therefore, at global scale, the influence of VPD can be higher than temperature. Similar result can be obtained from other models. Yuan et al. (2019) quantified the impacts of VPD and temperature on long-term changes of GPP using two satellite-based models (EC-LUE and MODIS model). For the EC-LUE model, global GPP decreased by 13.82 ± 3.12 Pg C with an increased VPD of 0.1 kPa and global GPP increased by about 6 Pg C with an increased temperature of 1℃. For the MODIS model, global GPP decreased by 18.29 ± 3.65 Pg C with an increased VPD of 0.1 kPa and global GPP increased by about 1 Pg C with an increased temperature of 1℃.

**7. Some letters in this paper look weird: 'v', 'x', 'z' and '%' are extremely small comparing other letters, please check the format before upload.**

Thank you, we checked these formats.

[revised manuscript text omitted]

---

## Author Response (AR4)

**Journal:** ESSD

**Title:** Improved estimate of global gross primary production for reproducing its long-term variation, 1982–2017

**MS No.:** essd-2019-126

**MS Type:** Data description paper

Dear Prof. Yuyu Zhou,

We appreciate the valuable opportunity to further revise our manuscript "Improved estimate of global gross primary production for reproducing its long-term variation, 1982–2017" (**MS No.:** essd-2019-126) for possible publication in ESSD.

We have carefully studied these comments and revised our manuscript accordingly. Please find attached the point-by-point responses to the comments of the reviewer. Please note that the comments from the reviewers are in **bold** followed by our responses in regular text. The changes in our manuscript are underlined with red.

We believe the quality of the manuscript can now meet the high standard of ESSD and deeply appreciate your consideration of our manuscript.

Sincerely,

Yi Zheng, Wenping Yuan

School of Atmospheric Sciences,

Sun Yat-sen University, Zhuhai 519082, Guangdong, China

Email: yuanwpcn@126.com

**Response to Reviewer:**

**The authors have answered the questions adequately. The revised EC-LUE model could work well at site level and for the global grids (the $R^2$ of 95% sites were larger than 0.5 and the gridded GPP had higher $R^2$ and lower RMSE than machine learning models and TRENDY products). Although the models still underestimated annual GPP significantly (slope<0.5 in Figure 3 and smaller daily GPP in Figure S2-S3), it fitted the tendencies well (Figure S1-S3) and performed better than the other models in this paper. It highlighted the usefulness to integrate $CO_2$ in the LUE models to simulate GPP. I think it's worth to accept this paper after following minor revision:**

Thanks for your comments. In our manuscript, we generated a long-term global gross primary production (GPP) dataset at 0.05° latitude by 0.05° longitude at 8-day interval using the revised EC-LUE model. The revised EC-LUE model performed well in simulating the spatial, seasonal, and interannual variations in GPP across the globe. Particularly, it has a unique superiority in reproducing the interannual variations in GPP compared with the original EC-LUE model and other LUE models by integrating the regulations of several major environmental variables with long-term changes, such as the atmospheric $CO_2$ concentration, radiation components, and vapor pressure deficit (VPD). The GPP dataset derived from the revised EC-LUE model provides an alternative and reliable estimate of global GPP at the long-term scale.

**1. Could you add the site R2/RMSE/Bias/tau of other models in figure 2 (can be the best one)?**

In Fig. 2, we examined the seasonal performance of our model by comparing against tower GPP at 8-day step. This study used 22 models, including satellite-based LUE models, machine learning models and biophysical models, in order to compare model performance, and especially to judge the improvements of revised EC-LUE model for identifying interannual variability of GPP. We appreciate your understanding that there are too many models to show into the Fig. 2. Therefore, we added a supplement figure similar to Fig. 2 to show the accuracy of our model and other models (or products) by comparing against tower GPP (Fig. S5). We added the corresponding result in the revised manuscript:

"Furthermore, the revised EC-LUE model had higher $R^2$, higher τ, lower RMSE, and lower absolute value of bias at most of the sites (Fig. S5)" (Line306-Line307 in the revised manuscript)

[Figure]

Figure S5: Site level comparisons between estimated GPP based on EC measurements and GPP simulations by the 22 models in Fig.4 at monthly step. This figure includes 8 LUE models (CASA, CFix, CFlux, MODIS, VPM, VPRM, EC-LUE, and the revised EC-LUE model) simulated using tower-derived meteorology data; the revised EC-LUE model, 3 machine learning models (FLUXCOM ANN, FLUXCOM MARS, and FLUXCOM RF), and 10 process-based biophysical models in TRENDY (CABLE, CLASS-CTEM, CLM, ISAM, JSBACH, JULES, LPJ-GUESS, LPX-Bern, ORCHIDEE, and VISIT) simulated using global reanalysis meteorology data.

**2. The mean $R^2$ of eight LUE models was mentioned but not determined in 3.1 and Figure S4.**

Perhaps the reviewer referred to the model mean $R^2$ we mentioned in Line294-Line306 (Section 3.1). In Section 3.1 and Fig. S4, we compared the $R^2$ of individual model (e.g., the revised EC-LUE model) and the mean $R^2$ of eight LUE models for each site, and we calculated the site percentage for an individual model (e.g., the revised EC-LUE model) with higher $R^2$ than the model mean $R^2$. That is, the "mean $R^2$" mentioned in 3.1 and Fig. S4 means "model mean $R^2$" value for each of the 95 sites, not a specific mean $R^2$ value.

Section 3.1 and Fig. S4 in our manuscript are as following:

Line294-Line306 (Section3.1): "For each site, we compared the $R^2$/RMSE/absolute value of bias of the individual model with the averaged value of all the eight LUE models (each site has an averaged $R^2$/RMSE/absolute value of bias) (Fig. S4a1-c1). The revised EC-LUE model had higher $R^2$ than the mean $R^2$ of the eight LUE models at 62% sites, which was comparable with the original EC-LUE model (63% sites) and VPM model (60% sites) (Fig. S4a1). Moreover, the revised EC-LUE model showed the lower RMSE and bias compared to mean values of all eight LUE models at 68% and 67% sites respectively, which indicated the better performance compared to the other LUE models at most sites (Fig. S4b1-c1). By using the global reanalysis meteorology data, we compared the performance of the revised EC-LUE model with three existing machine learning model products and ten process-based model products in TRENDY (Fig. 4b). The overall $R^2$ of the revised EC-LUE model ($R^2 = 0.57$) was higher than that of other models ($R^2$ ranged from 0.02 to 0.54) (Fig. 4b). The revised EC-LUE model, FLUXCOM ANN, and FLUXCOM MARS had more sites (over 90%) with higher $R^2$ than the mean $R^2$ (Fig. S4a2). And the revised EC-LUE model, FLUXCOM MARS, and FLUXCOM RF showed the lower RMSE at more than 90% sites (Fig. S4b2). Compared to the other models, the revised EC-LUE model had highest site percentage (81%) with lower absolute value of bias (Fig. S4c2)."

[Figure]

Figure S4: Percentage of sites for each model in Fig.4 where (a1, a2) $R^2$ of individual model > averaged $R^2$ of all models, (b1, b2) RMSE of individual model < averaged RMSE of all models, and (c1, c2) RMSE of individual model < averaged absolute value of bias of all models. All the monthly GPP simulations in (a1)-(c1) derived from tower-derived meteorology data, and all the monthly GPP simulations in (a2)-(c2) derived from global reanalysis meteorology data.

[revised manuscript text omitted]